# Measurement report: Impact of emission control measures on environmental persistent free radicals and reactive oxygen species – A short-term case study in Beijing

Yuanyuan Qin[1], Xinghua Zhang[2], Wei Huang[3], Juanjuan Qin[1], Xiaoyu Hu[1], Yuxuan Cao[1], Tianyi Zhao[1,5], Yang Zhang[1,6], Jihua Tan[1], Ziyin Zhang[4], Xinming Wang[5], Zhenzhen Wang[7]

[1]College of Resources and Environment, University of Chinese Academy of Sciences, Beijing, 100049, China

[2]Sinopec (Dalian) Research Institute of Petroleum and Petrochemicals Co., Ltd, Dalian, 116045, China

[3]Institute of Environmental Reference Materials of Environmental Development Centre of Ministry of Ecology and Environment, Beijing, 100029, China

[4]Institute of Urban Meteorology, China Meteorological Administration, Beijing, 100089, China

[5]Guangzhou Institute of Geochemistry, Chinese Academy of Sciences, Guangzhou, 510640, China

[6]Beijing Yanshan Earth Critical Zone National Research Station, Beijing, 101408, China

[7]School of Environmental Engineering, Changsha Environmental Protection Vocational College, Changsha, 410004, China

*Correspondence to*: Yang Zhang (zhangyang@ucas.ac.cn) and Jihua Tan (tanjh@ucas.ac.cn)

**Abstract.** A series of emission control measures implemented by the Chinese government have effectively reduced air pollution of multiple pollutants in many regions of the country in recent decades. However, the impacts of these control measures on environmental persistent free radicals (EPFRs) and reactive oxygen species (ROS), the two groups of chemical species that are known to be linked with adverse human health effects, are still not clear. In this study, we investigated the levels, patterns, and sources of EPFRs and gas- and particle-phase ROS (referred to as G-ROS and P-ROS, respectively) in Beijing during the 2015 China Victory Day Parade period when short-term air quality control measures were imposed. EPFRs in the non-control period (NCP) tended to be radicals centered on a mixture of carbon and oxygen, while those in the control period (CP) were mainly oxygen-centered free radicals. The contribution of G-ROS to the atmospheric oxidizing capacity increased or that of P-ROS decreased during CP compared to NCP. The strict control measures reduced ambient EPFRs, G-ROS, and P-ROS by 18.3%, 24.1%, and 46.9%, respectively, which were smaller than the decreases of most other measured pollutants. Although particle matter-based air quality control measures have performed well in achieving "Parade Blue", it is difficult to simultaneously reduce the negative impacts of atmosphere on human health. The "Parade Blue" days were largely attributed to the dramatic reduction in secondary aerosols, which were also largely responsible for EPFRs and ROS reductions. The source-sector based concentrations of $PM_{2.5}$, EPFRs, G-ROS, and P-ROS during CP were reduced by 78.7%–80.8% from secondary aerosols, 59.3%–65.0% from dust sources, 65.3%–67.0% from industrial emissions, and 32.6%–43.8% from vehicle emissions, compared to the cases during NCP. Furthermore, vehicle emissions and other inadequately controlled pollution sources may play a more complex role than expected in air quality and public health. This insight will prompt policymakers to reevaluate current air quality management strategies to more effectively address the challenges posed by pollutants such as EPFRs and ROS.

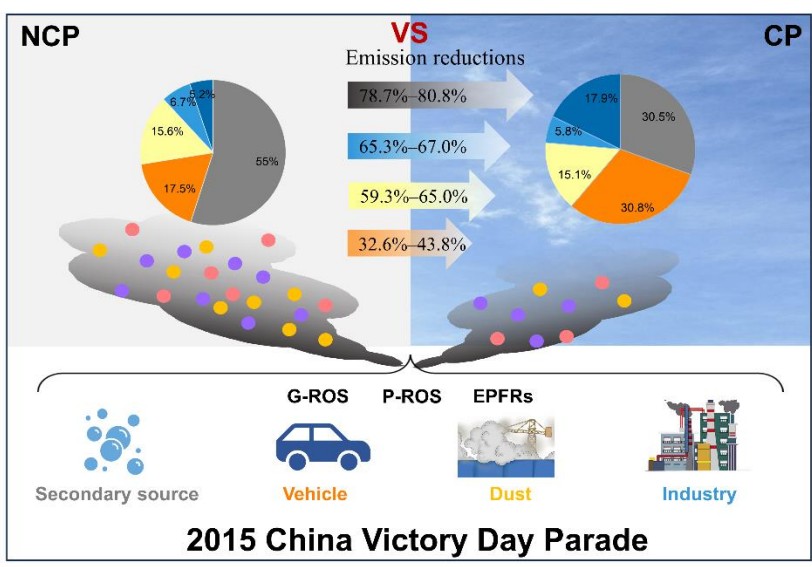

**Graphical abstract**

## 1 Introduction

Free radicals are atoms or molecules that contain at least one unpaired electron, which enables free radicals to be highly reactive (Khan et al., 2018). Free radicals attached to particles with a lifetime of several days or longer are defined as environmental persistent free radicals (EPFRs, e.g., phenoxy and semiquinone free radicals), to distinguish from traditional free radicals with a shorter lifetime (Li et al., 2022). The excessive lifetime of EPFRs will lead to a higher risk of human exposure to this group of chemical pollutants (Vejerano et al., 2018). It was estimated that human exposure to EPFRs in Beijing

is equivalent to approximately 33 cigarettes tar EPFRs inhaled per day (Xu et al., 2020). Numerous toxicological studies have shown that inhalation of EPFRs is linked to a variety of diseases, such as chronic lung disease and respiratory dysfunction, and thus has detrimental effects on human health (Chen et al., 2019b; Thevenot et al., 2013; Vejerano et al., 2018).

  Previous studies have shown that the concentration of EPFRs in atmospheric particles varied from $1.60 \times 10^{13}$ spins/m$^3$ to $8.97 \times 10^{15}$ spins/m$^3$ (Wang et al., 2022; Li et al., 2022). EPFRs are primarily derived from all most incomplete combustion

sources such as vehicle exhaust, biomass burning, and coal combustion (Wang et al., 2019b; Dugas et al., 2016; Saravia et al., 2013). EPFRs can be formed and stabilized on the surface of particulate matter containing transition metals and substituted aromatic structures emitted during combustion processes (Odinga et al., 2020; Chen et al., 2019a). For example, The incomplete combustion of vehicle emissions has been identified as an important source of EPFRs (Chen et al., 2018b). Dellinger et al. (2001) have shown that EPFRs in $PM_{2.5}$ are associated with combustion sources. Fang et al. (2023) found that

high concentrations of EPFRs are emitted from biomass burning. In addition to the combustion sources, EPFRs can also result from secondary processes in the atmosphere. It has been reported that EPFRs can be formed by the heterogeneous reaction of $O_3$ and polycyclic aromatic compounds (Borrowman et al., 2016a). EPFRs can also be formed from polycyclic aromatic hydrocarbons (PAHs) after photolysis (Li et al., 2022). Moreover, a recent study shows that EPFRs may also derive from dust sources (Li et al., 2023). Chen et al. (2018a) found that dust storms can increase the concentration of EPFRs in $PM_{2.5}$, and

metal oxides contained within dust particles provide the prerequisite conditions for EPFRs formation. Notably, EPFRs have received widespread attention in recent years because of their ability to convert $O_2$ molecules into reactive oxygen species (ROS) (Gehling et al., 2014). However, the sources and formation processes of EPFRs and ROS and the relationship between these two groups of pollutants are poorly understood, resulting in larger uncertainties in environmental risks assessments.

  ROS are oxygen molecules that contain at least one unpaired electron, including singlet oxygen, superoxide radicals ($O_2^{\bullet-}$),

OH radicals ($OH^{\bullet}$), hydrogen peroxide ($H_2O_2$), as well as organic radicals (Tong et al., 2018b; Arangio et al., 2016). Multiple sources of ROS have been identified, including wood combustion (Zhou et al., 2018), vehicle exhaust (Verma et al., 2010), and cooking emissions (Wang et al., 2020a). In addition, many studies have demonstrated that secondary sources related to photochemical reactions and oxidation reactions may be an important source of ROS. Volatile organic compounds (VOCs) and NOx have been shown to produce ROS through photochemical reactions (Venkatachari et al., 2007). ROS can also form on

the surface of particles or in air through reactions with ozone ($O_3$) under dark conditions (Zhu et al., 2018). $H_2O_2$ in secondary organic aerosols (SOA) has been found to be as much or more than ambient particles (Wang et al., 2012). $OH^{\bullet}$ and organic radicals can be formed from isoprene and β-pinene SOA, whereas $H_2O_2$ and $O_2^{\bullet-}$ mainly from naphthalene SOA has also been proposed (Tong et al., 2018a; Wei et al., 2021). These ROS play an active role in the atmospheric environment and determine the oxygenation of atmospheric aerosols. More importantly, ROS can cause oxidative stress, resulting in particle-related health

effects (Huang et al., 2018b). Oxidative stress, referred to as a state of disequilibrium between oxidizing agents (ROS) and antioxidant defense capacity, has been recognized as a major contributor to organism diseases (Fang et al., 2017). Thus, investigating the variations in the levels and sources of ROS are vital for understanding the mechanism of ROS formation and their effect on human health.

To mitigate air pollution and associated adverse health effects, the Chinese State Council issued a series of air quality control
plans since 2013, termed as the "Action plan on Prevention and Control of Air Pollution", which tremendously reduced the concentration of air pollutants in the following decade (Huang et al., 2018a; Niu et al., 2022). In addition, the Chinese government has implemented stricter short-term control measures to ensure excellent air quality during certain special periods such as when hosting mega-events (Wang et al., 2019a; Schleicher et al., 2012). In September 2015, the China Victory Day Parade was held in Beijing, and different levels of short-term emission measures were implemented in Beijing and surrounding
cities (Ma et al., 2020). Particle concentrations in Beijing were substantially reduced during this period, achieving the so-called "Parade Blue" (Huang et al., 2018b). Other air pollutants, such as primary organic aerosols (POA), SOA, water-soluble ions, and gaseous pollutants also exhibited significant reductions during this period (Zhao et al., 2017; Wang et al., 2017), demonstrating the potential of short-term control measures in reducing air pollution. However, the potential impacts of these measures on public health, especially regarding EPFRs and ROS, remain unclear. This event also provided an excellent
opportunity to quantify the effectiveness of control measures on EPFRs and ROS.

In this work, we evaluated the temporal variations in the chemical compositions of $PM_{2.5}$ and gas pollutants during the period when the 2015 China Victory Day Parade was held in Beijing, aiming to explore the influence of short-term air quality control measures on EPFRs, gas phase ROS (G-ROS), and particle phase ROS (P-ROS). Additionally, the sources and formation mechanisms of EPFRs, G-ROS, and P-ROS were explored using correlation analysis and positive factorization
matrix (PMF) model. The findings from this study have great implications for further understanding the sources and environmental risks of these chemical species and for the development of optimal air pollution control measures.

## 2 Methods and Materials

### 2.1 Sample Collection

All sampling were conducted on the rooftop of a five-floor building at the Institute of Remote Sensing and Digital Earth, Chinese Academy of Sciences ($117.39°E$, $40.01°N$), which is located between the fourth and fifth ring road in northern Chaoyang District, Beijing, China and is surrounded by residential buildings and Olympic Forest Park. A total of 76 $PM_{2.5}$ samples including 38 daytime (8:00–20:00) and 38 nighttime (20:00–8:00 the next day) samples were collected on prebaked quartz filters using Digital high-flow sampler (DHA-80, Digital, Switzerland) with a flow rate of 500 L/min from August 13 to September 19, 2015. The samples were wrapped in aluminum foil and then stored in a refrigerator at -20°C until analysis. Real-time $SO_2$, $NO_2$, and $O_3$ concentrations were simultaneously monitored online by an $SO_2$ analyzer (Model 43i, Thermo Scientific, USA), NOx analyzer (Model 42i, Thermo Scientific, USA), and ozone analyzer (Model 49i, Thermo Scientific, USA), respectively.

The specific sample information is shown in Table S1. The whole sampling period is divided into four sub-periods for analysis, with the specific control measures for each sub-period presented in Table S2. Period 1 (August 13–19) and period 4 (September 4–19) having no control measures implemented (referred to as non-control periods, NCP), period 2 (August 20–31) having regularly control measures, and period 3 (September 1–3) having stricter control measures, and periods 2 and 3 were defined as control periods (CP).

### 2.2 Chemical Analyses

Organic carbon (OC) and elemental carbon (EC) in $PM_{2.5}$ were measured by a thermal/optical carbon analyzer (model RT-4, Sunset Laboratory Inc. USA). Water-soluble ions ($NO_3^-$, $SO_4^{2-}$, $Cl^-$, $NH_4^+$, $Na^+$, $K^+$, $Ca^{2+}$, and $Mg^{2+}$) were analyzed by an ion chromatography analyzer (model ICS-1100, Thermo Fisher Scientific). Elements (Li, Na, Mg, Al, K, Ca, V, Mn, Fe, Co, Cu, Zn, As, Se, Rb, Cd, Pb, and Bi) in $PM_{2.5}$ were extracted by microwave digestion with 7 mL of ultrapure water, 2 mL of $HNO_3$, and 1 mL of $H_2O_2$, and the concentrations of elements were detected using inductively coupled plasma-mass spectrometry (ICP-MS). PAHs were extracted by a liquid mixture of dichloromethane and methyl alcohol and measured using gas chromatography equipped with a mass selective detector (Agilent 6890/5973 GC/MSD).

## 2.3 EPFRs Analyses

A 28×5 mm sampled filter was cut and placed in an electronic paramagnetic resonance (EPR) spectrometer (EMX plus, Bruker, Germany) to determine the concentrations of EPFRs. The measurement parameters of the EPR spectrometer were set as follows: the magnetic field strength was 3300–3450 G; the scanning time was 60 s; the microwave power was 8.0 mW; and the modulation amplitude was 2 G. The absolute spin amount and g factor were calibrated with $Mg^{2+}$ and $Cr^{3+}$ standards. Both of these standards have been proven effective for calibrating the g-factor and absolute spin number of EPFRs (Chen et al., 2019a; Chen et al., 2019b). During the calibration process, the $Mg^{2+}$ and $Cr^{3+}$ standard samples were inserted into the resonator and the system was tuned. The field offset was set to zero to ensure that the signal measured by the instrument matches exactly the signal for $Mg^{2+}$ and $Cr^{3+}$. The total spin numbers were divided by the volume of the samples, such that the concentration of EPFRs was expressed as spins/$m^3$. The crucial parameters for characterizing the type and abundance of EPFRs, such as the g-factor and line width ($\Delta H_{p\text{-}p}$), were extracted from the EPR spectrum. EPFRs with g-factor less than 2.003 are attributed to carbon-centered free radicals, such as cyclopentadienyl radicals, while EPFRs with g-factor of 2.004 and above are designated as oxygen-centered free radicals, such as semiquinone radicals (Zhu et al., 2019). Notably, semiquinone radicals have a resonance structure and can have an unpaired electron on the carbon atom. EPFRs with g-factor in the range of 2.003–2.004 suggested the existence of complex radicals centered on a mixture of carbon and oxygen or carbon-centered radicals containing oxygen atoms, such as phenoxy radicals (Yang et al., 2017; Hu et al., 2022).

## 2.4 G-ROS and P-ROS Measurements

The gas and aerosol collector-ROS (GAC-ROS) online monitoring system was used to measure the concentrations of G-ROS and P-ROS. The theory and constitutions of GAC-ROS were described in detail by Huang et al. (2016). In brief, GAC-ROS consists of a sampling section, a reaction and transportation section, and a detection section. Firstly, aerosols with aerodynamic diameter larger than 2.5 μm were removed by cyclone separator, gas was collected on the water film on the surface of the continuously rotating diffusion tube of the GAC, and $PM_{2.5}$ was trapped by supersaturated water vapor at a certain temperature. Secondly, solutions containing gas and particle samples were reacted with 2′,7′-dichlorofluorescin (DCFH) in the presence of horseradish peroxidase (HRP) in two glass reactors, respectively. The DCFH method has the lowest specificity and selectivity

for different types of ROS, capable of reacting with multiple ROS, including $H_2O_2$, as well as other short-lived ROS, such as OH radicals, superoxide radicals, peroxyl radicals, and peroxynitrite (Bates et al., 2019). Finally, a fluorescence detector was used to measure the concentrations of G-ROS and P-ROS. For data accuracy, fresh DCFH and HRP were prepared at least every two days, and $H_2O_2$ standard curves were created daily.

**2.5 Source Apportionment**

Researchers have successfully employed PMF for source apportionment o EPFRs and ROS (Ainur et al., 2023; Wang et al., 2019b). In this study, we used the Environmental Protection Agency (EPA) PMF 5.0 version to perform the source apportionment of $PM_{2.5}$, EPFRs, G-ROS, and P-ROS. The fundamental principle of PMF involves first calculating the errors of various chemical components in particulate matter using weights, followed by utilizing the least squares method to estimate the main pollution sources of the particulate matter and their contribution. The PMF model decomposes a matrix of specific sample data (X) into a source contributions matrix (G) and factor profile matrix (F), as well as a residual matrix (E), as shown in the following equation:

$$X_{ij} = \sum_{k=1}^{p} g_{ik} f_{kj} + e_{ij} \tag{1}$$

Where $X_{ij}$ denotes the concentration of the $j$th species in the $i$th sample, $g_{ik}$ represents the source contribution of the $k$th factor to the $i$th sample, $f_{kj}$ is the factor profile of $j$th species in the $k$th factor, and $e_{ij}$ is the residual matrix.

PM$_{2.5}$, EPFRs, G-ROS, P-ROS, OC, EC, water-soluble ions ($NO_3^-$, $SO_4^{2-}$, $Cl^-$, $NH_4^+$, $Na^+$, $K^+$, $Ca^{2+}$, and $Mg^{2+}$), and elements (Na, Mg, Al, K, Ca, V, Mn, Fe, Co, Cu, Zn, As, Se, Rb, Cd, Pb, and Bi) were included in the PMF model with a total sample number of 76. The procedure for the PMF model has been described in many previous reports (Wang et al., 2019b; Sharma et al., 2016). Missing concentration values were replaced with "-999". The component concentration was changed to half of the method detection limit (MDL) when it was lower than the MDL. The calculation formula of uncertainty is Uncertainty=K×C, where K is the analytical uncertainty and C represents the concentrations of the chemical components. The quality of the data was evaluated according to the signal-to-noise ratio (S/N), and species with S/N ranging from 1 to 10 were categorized as "Strong", while those with S/N ranging from 0.5 to 1 were categorized as "Weak". The tracer species were also categorized as "Strong". The degree of rotation in the model results was controlled by the FPEAK and FKEY values.

**3 Results and Discussion**

**3.1 Temporal Variations of Air Pollution**

To investigate the effectiveness of short-term air pollution control measures on pollutants concentrations during the 2015 China Victory Day Parade, temporal variations of $PM_{2.5}$, EC, middle molar weight PAHs (MMW-PAHs, 4 ring PAHs), elements, and gas pollutants were first examined. As shown in Figure 1, $PM_{2.5}$ concentration decreased continuously from period 1 to period 3 before rebounded in period 4, with average $PM_{2.5}$ concentration being ~60% lower in CP (periods 2 and 3) than NCP (periods 1 and 4). Similarly, EC, a typical marker of fossil fuel combustion (Zhang et al., 2015; Wang et al., 2020b), was also ~57.0% lower in CP than NCP, demonstrating that provisional control measures have significantly reduced fossil fuel combustion emissions. Additionally, a 32% lower MMW-PAHs concentration in CP than NCP implied that the control measures were also effective in reducing emissions from diesel vehicle exhaust (Perrone et al., 2014). The concentrations of elements also decreased dramatically with a 51.4% lower concentration in CP than NCP.

Regarding the gaseous pollutants, the concentrations of $O_3$, $SO_2$, and $NO_2$ decreased by 10.8%, 51.2%, and 45.5%, respectively, during CP compared to those in the NCP. $NO_2$ is mainly derived from vehicle exhaust emissions, and $SO_2$ is mainly from the fossil fuel (e.g., coal) combustion (Hien et al., 2014; Ma et al., 2020). Apparently, the control measures implemented during CP have effectively reduced emissions from industrial coal combustion and vehicle exhaust, both of which are important combustion sources. In contrast, the reduction in $O_3$ during the CP was much less than that of $NO_2$, which can be explained by the reduction in the titration reaction between $O_3$ and NO due to the reduced NO emission from vehicle exhaust (Guo et al., 2016; Okuda et al., 2011). Moreover, these results showed that the percentage decrease in gaseous pollutants were smaller than in $PM_{2.5}$.

Different diurnal variations were observed between the pollutants. The average concentrations of EC and $NO_2$ were generally higher during the nighttime (1.33 μg/m³ and 40.2 μg/m³, respectively) than daytime (0.82 μg/m³ and 28.4 μg/m³, respectively) in the whole measurement period. This is especially the case during the NCP, likely due to increased nighttime traffic emissions or the occurrence of temperature inversions (Yang et al., 2015; Wu et al., 2012). During daytime, the restrictions on heavy-duty vehicles entering the urban areas of Beijing may lead to increased emissions from diesel vehicles near the fourth and fifth

ring roads at nighttime (Cai et al., 2020). Similar diurnal variations have also been observed previously in Agra and Beijing (Pipal et al., 2014; Lin et al., 2009; Ke et al., 2017; Cai et al., 2020). Additionally, lower temperatures and reduced solar radiation at nighttime decrease the photolysis of $NO_2$, which is the main chemical mechanism for $NO_2$ loss at daytime (Cai and Xie, 2010), further contributing to the elevated $NO_2$ concentrations at nighttime. $O_3$ was higher in the daytime than nighttime, indicating intensive photochemical actions. $SO_2$ concentration significantly increased in the daytime during period 4, which may be caused by the surge in industrial activities (He et al., 2017).

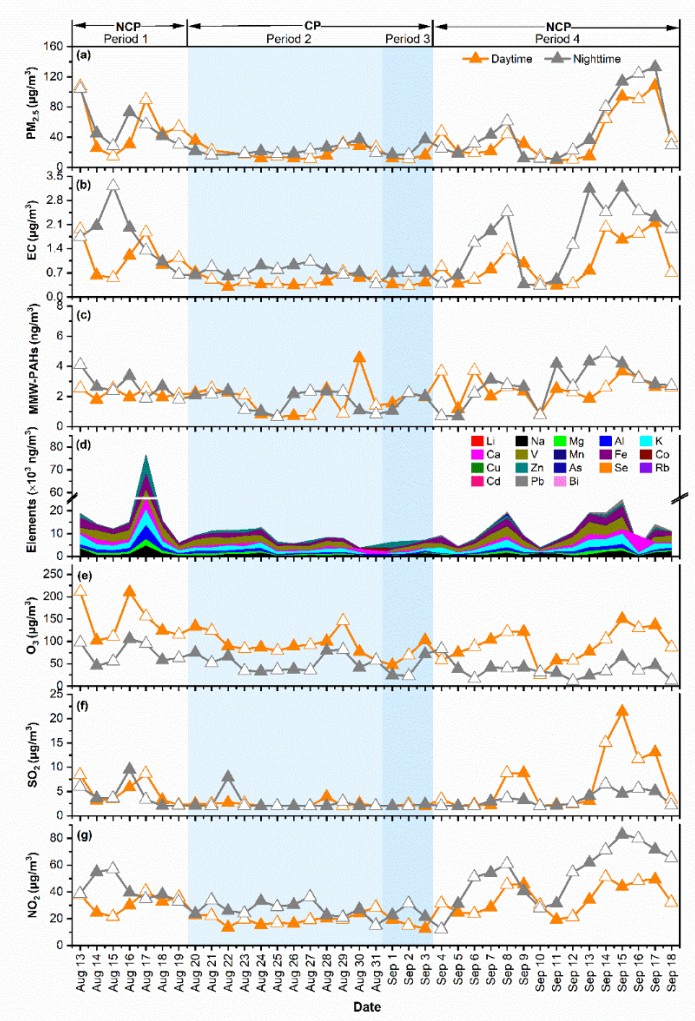

**Figure 1: Temporal variations in concentrations of (a) PM$_{2.5}$, (b) EC, (c) MMW-PAHs, (d) elements, (e) O$_3$, (f) SO$_2$, and (g) NO$_2$ during the four sub-periods of the 2015 China Victory Day Parade.**

## 3.2 Characteristics of Environmentally Persistent Free Radicals (EPFRs)

Figure 2 shows the temporal variations in EPFRs concentrations during the whole measurement period. The average concentration of EPFRs was $(1.00\pm0.75)\times10^{14}$ spins/m$^3$ during NCP and $(8.19\pm5.60)\times10^{13}$ spins/m$^3$ during CP, which represents 18.1% lower concentration during CP than NCP. The percentage decrease in EPFRs was smaller than most of the other measured pollutants (PM$_{2.5}$, EC, elements, NO$_2$, and SO$_2$). Notably, the concentration of EPFRs increased during period 3, despite the reduction in emission intensity from pollution sources under strict control measure conditions, suggesting the impact on the formation of EPFRs from certain characteristic sources still be modest. Chen et al.(2020) showed that the change in the EPFRs concentrations is unrelated to the change in the PM concentration, but rather determined by their source characteristics. For instance, activities during the parade may have increased the contribution from traffic and other sources. Detailed discussion on these source characteristics will follow in subsequent source apportionment sections. Nevertheless, the levels of EPFRs in PM$_{2.5}$ in this study were approximately two orders of magnitude lower than those in Beijing ($1.70\times10^{15}$–$3.50\times10^{16}$ spins/m$^3$) in 2016 (Yang et al., 2017) and slightly lower than those in Xi'an ($1.79\times10^{14}$ spins/m$^3$) in 2017 (Wang et al., 2019b), but much higher than that in Chongqing ($7.0\times10^{13}$ spins/m$^3$) in 2017–2018 (Qian et al., 2020).

The average concentrations of EPFRs during the daytime and nighttime were $6.85\times10^{13}$ spins/m$^3$ and $1.18\times10^{14}$ spins/m$^3$, respectively, indicating that nighttime samples contained more EPFRs than daylight samples. The lower EPFRs concentrations during daytime may be related to the rapid conversion of EPFRs to other chemical species under strong irradiation. Previous studies have shown that the half-life times of EPFRs were shorter under light than dark conditions (Lang et al., 2022; Chen et al., 2019a), suggesting that light irradiation promotes the transformation of EPFRs (Jia et al., 2019). For instance, semiquinone radicals can rapidly degrade into $CO_2$ under light irradiation conditions (Li et al., 2014). Besides, increased traffic emissions during nighttime, as mentioned above, may have possibly led to higher levels of EPFRs in nighttime. For instance, Hwang et al.(2021) found that PM$_{2.5}$ from traffic-related sources generally has higher EPFRs concentrations than that from urban background particles.

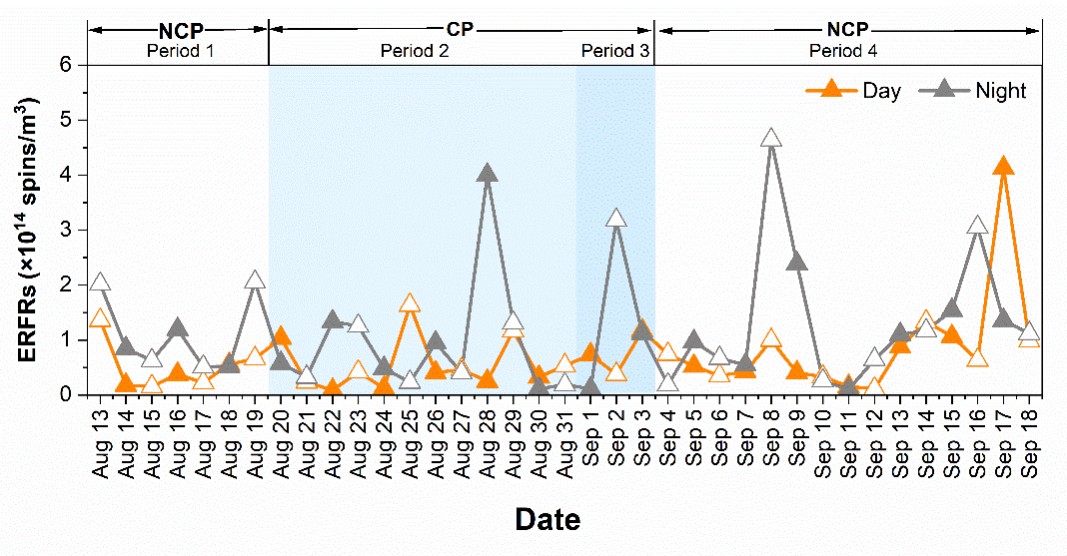

**Figure 2: Temporal variations in EPFRs concentrations during the measurement period.**

The g-factor and $\Delta H_{p-p}$ values of EPFRs during the four pollution periods are depicted in Figure 3. The average g-factor was

240 2.00395 in NCP and 2.00429 in CP. Hence, the observed EPFRs in NCP tend to be radical centered on a mixture of carbon and oxygen. The higher g-factor in CP, especially in period 3, suggested that oxygen-centered free radicals were attached to the $PM_{2.5}$ samples (Li et al., 2023). It has been reported in literature that EPFRs derived from primary combustion sources (e.g., coal combustion and vehicle emission) generally have a lower g-factor (Chen et al., 2019c). The data presented above indicated that the generation of EPFRs with lower g-factor was decreased during CP when the emissions from combustion sources were significantly reduced. It is known that carbon-centered radicals are more unstable and easily oxidized in the atmosphere than

245 oxygen-centered radicals (Wang et al., 2018). Therefore, the free radicals generated during CP were less susceptible to further oxidation, while those generated during NCP were more easily oxidized. The average $\Delta H_{p-p}$ of EPFRs during CP and NCP was 4.62 ± 1.06 G and 4.42 ± 0.87 G, respectively. The slightly larger $\Delta H_{p-p}$ during CP than NCP indicates a relatively complex path for the formation of EPFRs under strict control measure conditions. This may be explained by a marked increase in the activity of other sources, which will be discussed below. However, due to the complex formation and transformation of EPFRs,

current evidence does not sufficiently explain the changes in EPFRs, necessitating further investigation to uncover deeper

mechanisms.

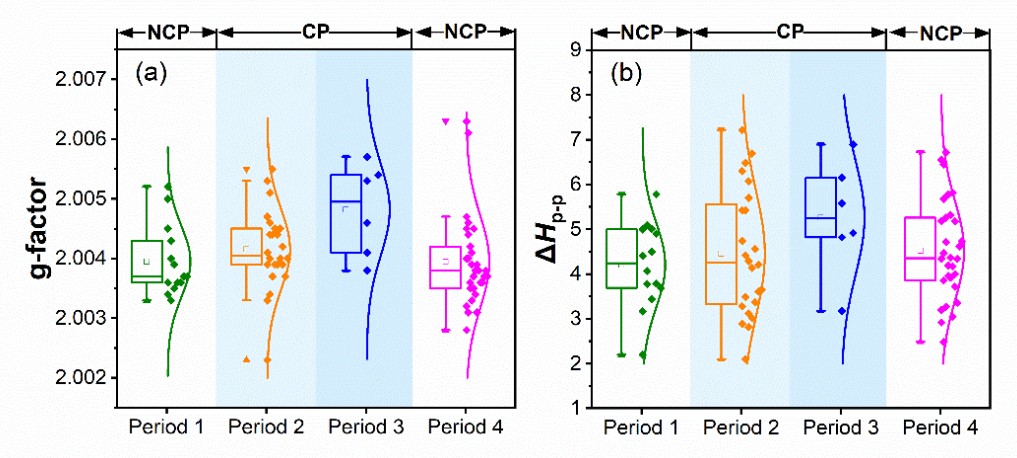

**Figure 3: The (a) g-factor and (b) $\Delta H_{p-p}$ of EPFRs during the four pollution periods.**

### 3.3 Characteristics of Reactive Oxygen Species (ROS) Activity

The ROS activity obtained here was expressed in nmol $H_2O_2$ equivalents $m^{-3}$. As shown in Figure 4, the average concentrations

of G-ROS and P-ROS were 17.2±2.51 nmol $H_2O_2/m^3$ and 13.6±2.71 nmol $H_2O_2/m^3$, respectively, during NCP, decreased to

13.8±1.29 nmol $H_2O_2/m^3$ and 7.25±1.79 nmol $H_2O_2/m^3$ during period 2, and further decreased to 10.3±0.63 nmol $H_2O_2/m^3$ and

7.02±0.57 nmol $H_2O_2/m^3$ during period 3. The concentrations of ROS during CP were comparable to those observed in urban

America (Wang et al., 2011) and rural China (Zhao et al., 2023; Huang et al., 2016). It is noteworthy that the impact of the

control measures on G-ROS and P-ROS was different. Compared with NCP, the percentage decrease in G-ROS during CP was

24.1%, which was lower than that P-ROS decrease of 46.9%. This difference may be related to the complex formation and

transformation mechanism of G-ROS. These results further suggested that the decrease in gaseous pollutants was lower than

that in particulate pollutants. Furthermore, the much higher ratios of G-ROS to P-ROS during CP than NCP suggested that the

contribution of G-ROS to atmospheric oxidizing capacity was increased or that of P-ROS was decreased during this period

(Figure 5). However, the control measures were clearly less effective in reducing G-ROS during period 2 than period 3.

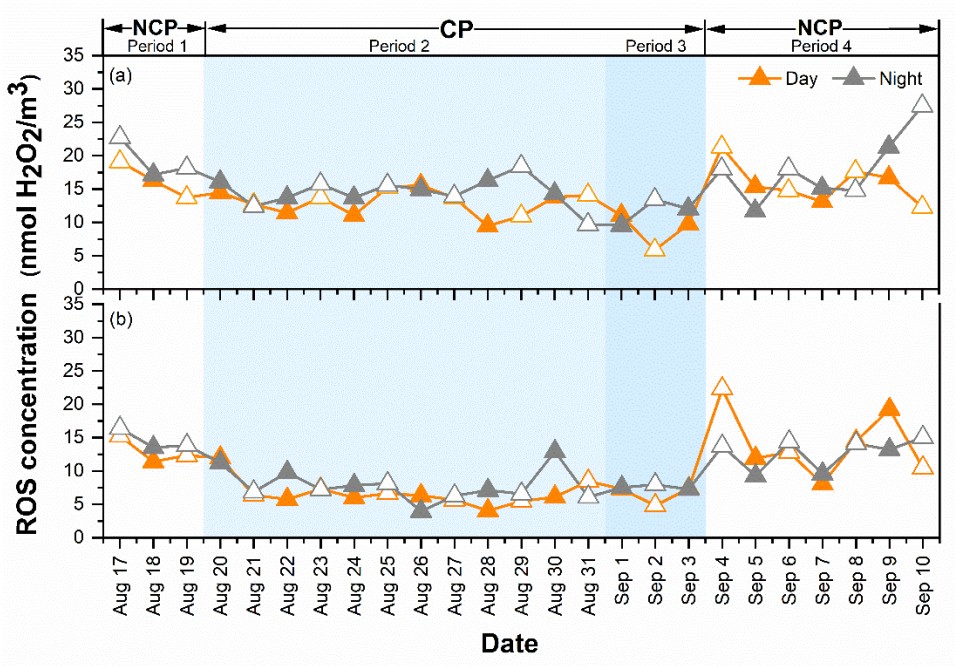

**Figure 4: The concentrations of (a) G-ROS and (b) P-GOS during the whole measurement period.**

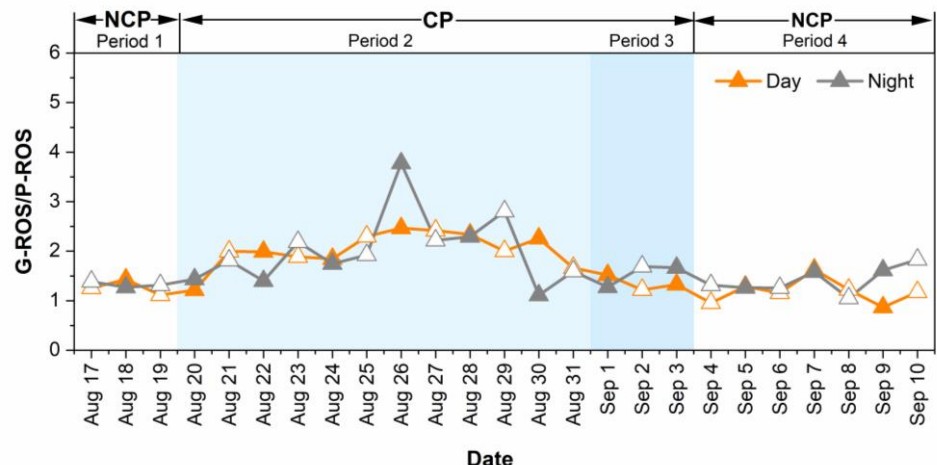

**Figure 5: The ratios of G-ROS to P-ROS during the whole measurement period.**

The average concentration of G-ROS was slightly higher at nighttime (15.8 nmol $H_2O_2/m^3$) than daytime (13.7 nmol $H_2O_2/m^3$), so was the case of P-ROS (10.0 nmol $H_2O_2/m^3$ versus 9.5 nmol $H_2O_2/m^3$), consistent with that reported in a previous study in

Xi'an in 2021 (Ainur et al., 2023). The higher ROS levels at night are more evident from the diurnal variations shown in Figure 6. G-ROS decreased at approximately 8:00 am and then rapidly increased at 18:00 pm during all of the four sub-periods. However, P-ROS decreased at approximately 3:00 am and then increased at approximately 13:00 pm. These results suggest that different formation mechanisms existed between G-ROS and P-ROS.

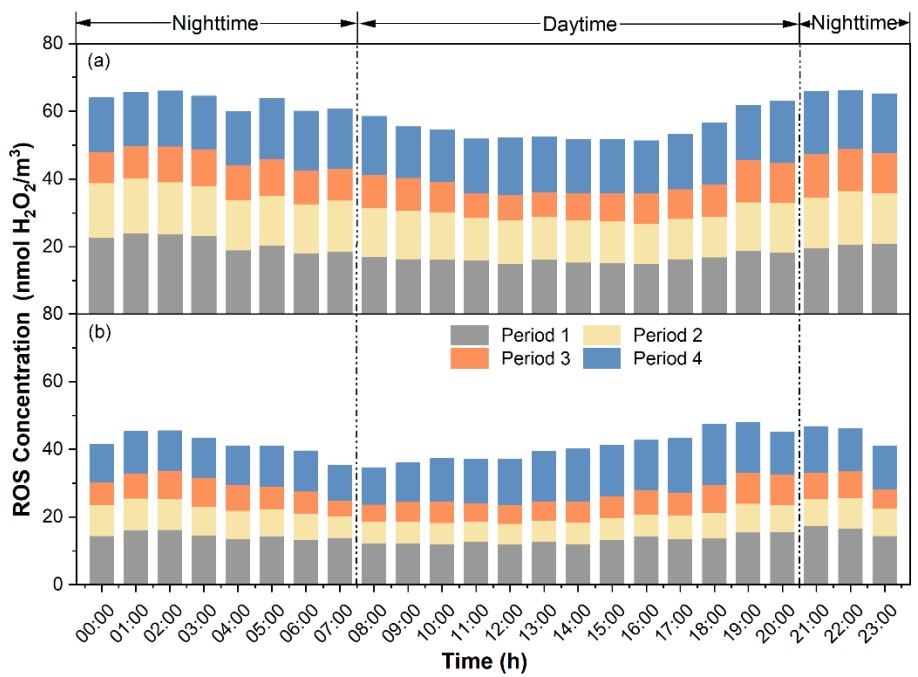

Figure 6: Diurnal variations in concentrations of (a) G-ROS and (b) P-ROS during the four pollution periods.

**3.4 Correlation Analysis**

Figure 7 shows the Spearman correlation of EPFRs, G-ROS, and P-ROS with other pollutants. EPFRs concentration strongly correlated with $\Delta H_{p\text{-}p}$ ($r>0.76$), indicating more abundant types of EPFRs under higher EPFR concentration condition. In addition to simple and well-defined EPFRs, such as semiquinone and cyclopentadienyl radicals, there are also many complex and unknown EPFRs in the research blind spot. AQI and $PM_{2.5}$ both positively correlated with the concentration of EPFRs ($r>0.42$), suggesting that the contamination of EPFRs was significantly influenced by the relevant health index and haze. EPFRs exhibited a significant positive correlation with vehicle exhaust markers EC and $NO_2$ ($p<0.05$), emphasizing that

vehicle exhaust emissions may be an important source of EPFRs in Beijing. Recent studies also found that EPFRs significantly correlated with EC and $NO_2$ on highways, mainly related to the emissions in vehicle exhaust (Hwang et al., 2021; Li et al., 2022). A stronger positive correlation between ERFRs and secondary inorganic ions was found in the daytime ($r=0.45$) than nighttime ($r=0.37$). Meanwhile, a significant positive correlation between ERFRs and $O_3$ was also observed in the daytime ($p<0.1$), consistent with the results of Chen et al (2019b). The oxidation of different types of PAHs by $O_3$ could form different types of EPFRs, as demonstrated in a previous study (Borrowman et al., 2016b). However, Huang et al. observed a negative correlation between EPFRs and $O_3$ at highway sites and believed that this may be due to the consumption of $O_3$ by NO (Hwang et al., 2021). In this study, the hot summer conditions were more conducive to the conversion of PAHs into EPFRs, especially in the presence of high $O_3$ concentrations. This implies that the mechanism of EPFRs generation varies under different environmental conditions. Transition metals, as single-electron acceptors or shuttles (Wan et al., 2020), play a key role not only in the formation of EPFRs but also in maintaining the long half-life of EPFRs (Pan et al., 2019; Vinayak et al., 2022). Cd only significantly correlated with EPFRs in the daytime ($p<0.05$), while the majority of transition metals (e.g., Mn, Fe, V, and Cd) significantly correlated with EPFRs in the nighttime. These results suggest that EPFRs in the nighttime were stabilized in particles via transition metals from fuel combustion processes, while an increased proportion of EPFRs was generated via other pathways in the daytime, such as the secondary reactions mentioned above.

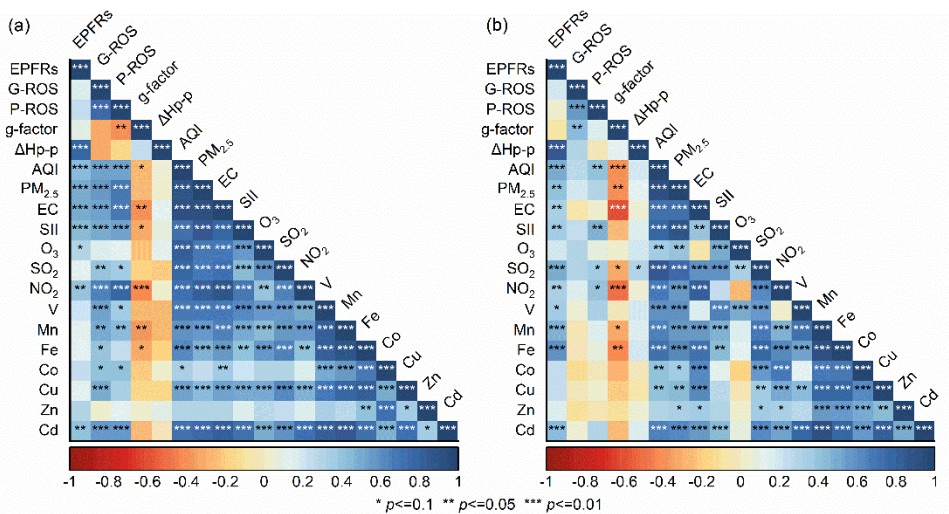

**Figure 7: Spearman correlation matrix of EPFRs, G-ROS, and P-ROS concentrations with meteorological parameters, gaseous pollutants, and PM$_{2.5}$ components during (a) daytime and (b) nighttime. SII: secondary inorganic ions. Red and blue color denote a**

 **negative and a positive correlation, respectively.**

It is not surprising that P-ROS associated with $PM_{2.5}$ more than G-ROS did. Both G-ROS and P-ROS strongly correlated with EC and $NO_2$ ($r>0.54$) in the daytime, suggesting that G-ROS and P-ROS were derived from traffic-related emissions, as was reported in a previous study (Stevanovic et al., 2019). G-ROS ($r>0.52$) and P-ROS ($r>0.54$) also correlated well with secondary inorganic ions in the daytime, indicating secondary aerosols as another important source of G-ROS and P-ROS. G-ROS significantly correlated with the majority of transition metals (e.g., V, Mn, Fe, Co, Cu, and Cd) ($p<0.1$), and P-ROS positively correlated with metals (e.g., V, Mn, Co, and Cd), which is consistent with the current knowledge regarding the metal-induced ROS formation mechanism. Transition metals have been considered to be capable of generating excess ROS such as hydroxyl and superoxide radicals via Fenton-like reactions (Brehmer et al., 2019; Lin and Yu, 2020).

During the nighttime, there were moderate correlations between P-ROS and AQI, $NO_2$, $SO_2$, and secondary inorganic ions, indicating the contributions of vehicle exhaust emissions, coal combustion emissions, and secondary formation to P-ROS. The very weak correlation of both G-ROS and P-ROS with $O_3$ implied limited formation of G-ROS and P-ROS from secondary reaction processes caused by $O_3$ in Beijing. The correlations of G-ROS and P-ROS with EPFRs were also very weak. Although EPFRs can induce the formation of single ROS species (e.g., OH· and $O_2\cdot^-$) (Hwang et al., 2021; Guo et al., 2020), individual ROS species is incapable of existing alone in the air, leading to different interactions between EPFRs and different ROS species.

## 3.5 Source Apportionment

The source profiles of $PM_{2.5}$ were analyzed using the PMF model. As shown in Figure 8, if considering the whole campaign together, five major source factors were identified. The high proportion of $NO_3^-$, $SO_4^{2-}$, and $NH_4^+$ are attributed to secondary aerosols. A factor is recognized as vehicle emissions due to the high abundance of EC and Cu. Another factor can be recognized as dust sources because of the high proportion of Mg, Al, Ca, and Fe. A fourth factor was linked to industrial emissions sources due to the high proportion of V, Mn, Rb, Cd, Pb, and Bi. Additionally, another factor was identified as other sources because of the high abundance of Co and Zn. Secondary aerosols, which accounted for the largest faction (52.0%), followed by vehicle emissions (20.8%), dust sources (13.5%), industrial emissions (6.3%), and other sources (7.4%), the total of which resolved

95.4% of PM$_{2.5}$. The percentage contributions from each source factor to PM$_{2.5}$ differed to some extent between NCP and CP (Figure 9). For example, the percentage contributions from the above five source factors were 55.0%, 17.5%, 15.6%, 6.70%, and 5.22%, respectively, during NCP (Figure S1), and were 30.5%, 30.8%, 15.1%, 5.77%, and 17.9%, respectively, during CP. The large decrease in percentage contribution form secondary aerosols during CP was due to the tremendous reductions in precursor gases (e.g., SO$_2$ and NO$_2$) of secondary aerosols. The percentage contribution from vehicle emissions actually increased because the concentration decrease from this sector was smaller than those from the other major source sectors (especially the factor of secondary aerosols, as discussed below). Further, the smaller decrease in concentrations from vehicle emissions may be attributed to the World Athletics Championships held at the National Stadium (known as the Bird's Nest) during period 2, which resulted in a significant increase in traffic flow near the sampling site. As a result, the percentage contribution from vehicle emissions increased significantly during CP.

The concentrations of PM$_{2.5}$ fractions from most source factors decreased during CP compared to NCP (Figure S2), e.g., by 78.7%, 32.6%, 63.0%, and 67.0%, from secondary aerosols, vehicle emissions, dust sources, and industrial emissions, respectively, due to the strict emission control measures implemented during CP. Thus, the achievement of "Parade Blue" days was largely attributed to the dramatic decreases in secondary aerosols, dust, and industrial emissions, a phenomenon that is consistent with that observed in a previous study during the Asia-Pacific Economic Cooperation conference reported by Sun et al. (2016). Obviously, the strict control measures during the parade period worked effectively in reducing both primary and secondary pollutants.

The predominant sources of EPFRs during NCP were also secondary aerosols (50.6%), followed by vehicle emissions (33.5%), other sources (9.89%), dust sources (4.12%), and industrial emissions (1.85%). The percentage contributions of these source sectors to EPFRs during CP changed to 20.8%, 43.7%, 31.3%, 3.0%, and 1.25%, respectively. Vehicle emissions surpassed secondary aerosols to become the largest sources of EPFRs during CP. Additionally, contributions from other sources also significantly increased during CP, especially during period 3. During NCP, secondary aerosols were also the largest source (45.9%) of G-ROS, followed by vehicle emissions (36.6%), dust sources (11.2%), other sources (5.78%), and industrial emissions (0.54%), respectively. During CP, the contribution of secondary aerosols decreased remarkably to 18.3%, while that of vehicle emissions increased significantly to 43.0%, as well as other sources increased significantly to 28.9%. Similarly, the

predominant source of P-ROS during NCP was also secondary aerosols (44.2%), followed by vehicle emissions (30.0%), dust sources (12.9%), other sources (10.0%), and industrial emissions (2.73%). During CP, the contribution of secondary aerosols (17.7%) to P-ROS dropped significantly while that of vehicle emissions and increased slightly to 35.2%, and other sources increased significantly to 35.9%. Although some pollutants have been effectively regulated during CP, the levels of harmful species such as EPFRs and ROS have not decreased significantly in sync. The PMF results imply that vehicle emissions and other inadequately controlled pollution sources (defined as other sources) may play a more complex role than expected in air quality and public health. This finding will prompt policymakers to reevaluate existing air quality control measures to more effectively address the challenges posed by pollutants such as EPFRs and ROS.

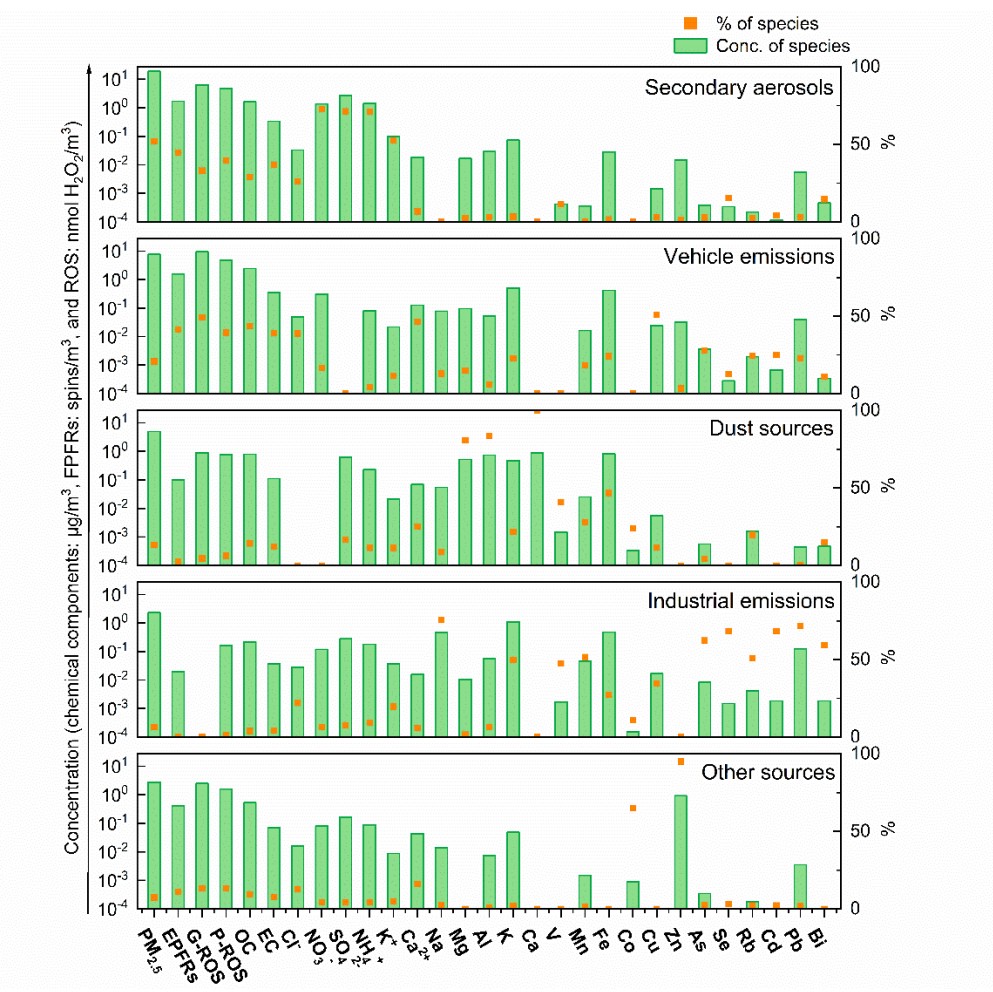

**Figure 8: Profiles of different source factors of PM$_{2.5}$, including EPFRs, G-ROS, and P-ROS.**

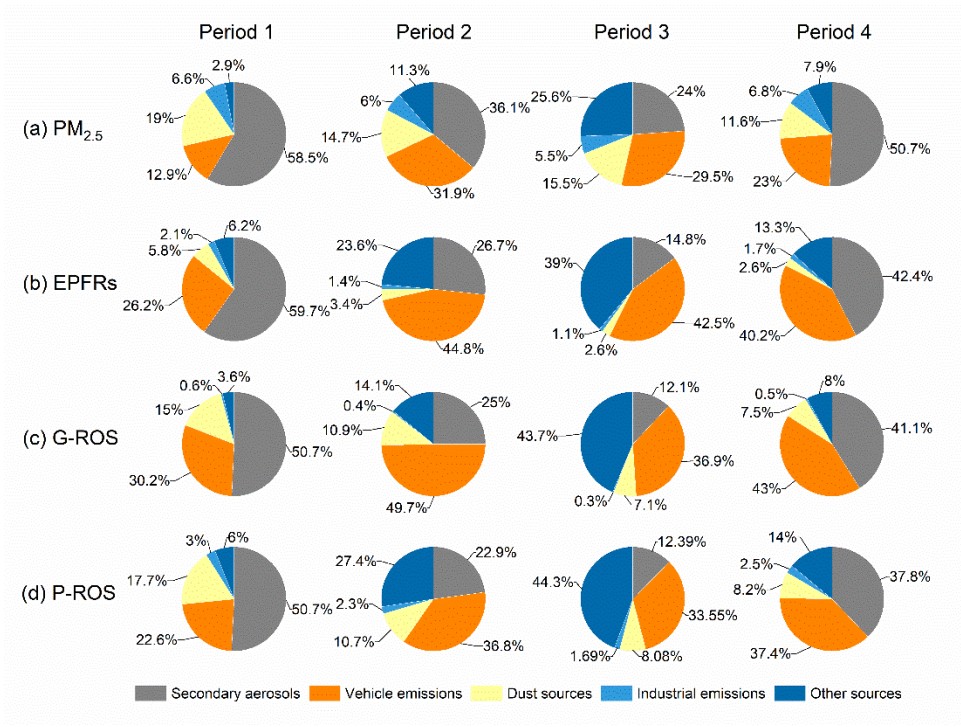

**Figure 9: The contributions of different sources to PM₂.₅, EPFRs, G-ROS, and P-ROS during the four sub-period.**

## 4 Conclusions

The short-term air quality control measures on hazardous substances during the 2015 China Victory Day Parade in Beijing reduced the concentrations of EPFRs by 18.3%, G-ROS by 24.1%, and P-ROS by 46.9% during CP compared to NCP. Overall,

the decrease in EPFRs and ROS was smaller than those for most other measured pollutants (e.g., PM₂.₅, EC, elements, and SO₂). Although particle matter-based air quality control measures have performed well in achieving "Parade Blue", it is difficult to simultaneously reduce the negative impacts of atmosphere on human health. Given that EPFRs and ROS exhibited a significant positive correlation ($p<0.01$) with EC, secondary inorganic ions, $NO_2$, and Cd, controlling emissions of these chemical species would benefit the reduction in EPFRs and ROS pollution. The sources of EPFRs and ROS differed between

375   day- and nighttime. EPFRs were mainly from vehicle exhaust emissions and atmospheric oxidation processes in the daytime and vehicle exhaust emissions and fossil fuel combustion in the nighttime. Vehicle exhaust, secondary aerosols, and metals

from fuel combustion processes were important sources of G-ROS and P-ROS in the daytime, while vehicle exhaust and coal combustion emissions were the major contributors of P-ROS in the nighttime. The predominant sources of $PM_{2.5}$, EPFRs, G-ROS, and P-ROS during NCP were secondary aerosols, followed by vehicle emissions, but vehicle emissions surpassed secondary aerosols to become the predominant source of these chemical species during CP. The control measures implemented during CP reduced source-sector based concentrations of $PM_{2.5}$, EPFRs, G-ROS, and P-ROS by 78.7%–80.8% from secondary aerosols, 59.3%–65.0% from dust sources, 65.3%–67.0% from industrial emissions, and 32.6%–43.8% from vehicle emissions, compared to the cases during NCP. Results from this study will benefit the development of future air quality management policies targeting EPFRs and ROS. However, the generation and transformation processes of EPFRs and ROS involve multiple complex chemical reactions, further in-depth studies are still needed to gain a complete understanding of the formation pathways of EPFRs and ROS under different environmental conditions. For example, it is necessary in future to conduct smog chamber or flow tube experiment to simulate the photochemical reactions and oxidation processes of EPFRs and ROS in the atmosphere, as well as their interactions with different chemical species. In addition, substantial efforts are needed to develop the pretreatment and analytical methods to separate various EPFRs and gain the detailed information such as g-tensors and hyperfine splitting constants to identify the specific structures of the radicals, consequently clarifying the link between EPFRs and ROS.

**Data availability.** The data used in this study are available on the Zenodo data repository platform: https://doi.org/10.5281/zenodo.10136894 (Qin et al., 2023).

**Author contribution.** YZ and JT: Conceptualization and Writing – review & editing. YQ: Writing - Original Draft and Writing - Review & Editing. XZ: Writing - Review & Editing. WH: Investigation. JQ: Methodology. XH: Software. TZ: Software. ZZ: Investigation. XW: Methodology. ZW: Funding acquisition.

**Competing interests.** The authors declare that they have no conflict of interest.

**Disclaimer.** Publisher's note: Copernicus Publications remains neutral with regard to jurisdictional claims in published maps

and institutional affiliations.

**Financial support.** This work was supported by the National Key Research and Development Program of China (2022YFC370300, 2023YFE0102400, 2022YFC3703402, and 2022YFC3701103), and the Provincial Natural Science Foundation of Hunan (2023JJ30004).

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
