# Peer review of "Measurement report: Impact of emission control measures on environmental persistent free radicals and reactive oxygen species $-\mathbf{A}$ short-term case study in Beijing"

_EGUsphere, 2023_

## Author Comment (AC1)

Response to reviewer:

We greatly appreciate the reviewer' thoughtful comments, suggestions, and recognition of the value and significance of our study. We have carefully addressed all the comments, and a detailed item-by-item response to the comments of the reviewer is given below. The original comments are in black and our responses are highlighted in blue. Additionally, we have included the revised paragraph following each response to demonstrate the changes made.

Comments:

The manuscript reports the impacts of control measures on environmentally persistent free radicals (EPFRs) and reactive oxygen species (ROS) in Beijing during the 2015 China Victory Day Parade period. The findings reveal the changes in sources and formation processes of EPFRs and ROS in different control period. The paper is well organized and the topic is interesting for the control of hazardous species. However, some issues should be modified before publication, specific comments are as follow:

1. Abstract: You should provide the variation of the main sources before and after CP period as a comparison for those during the CP period.

**Response: Thank you very much for your valuable advice.** According to your suggestion, we have added the variation of different sources during control period (CP) and non-control period (NCP) in Line 37–39, which are now described as follow:

"The "Parade Blue" days were largely attributed to the dramatic reduction in secondary aerosols, which were also largely responsible for EPFRs and ROS reductions. The source-sector based concentrations of $PM_{2.5}$, EPFRs, G-ROS, and P-ROS during CP were reduced by 78.7%–80.8% from secondary aerosols, 59.3%–65.0% from dust sources, 65.3%–67.0% from industrial emissions, and 32.6%–43.8% from vehicle emissions, compared to the cases during NCP."

2. Graphical abstract: The resolution of figure is too low.

**Response: We are extremely grateful for pointing out this problem.** We have

reworked graphical abstract to improve the resolution of figure are the same as follows:

[Figure]

**Graphical abstract**

3. Line 36: "correlations", between what?

**Response: Sorry for the confusion.** Actually, "correlations" here refers to the correlations between EPFRs and ROS. Considering the word-count limitation of the abstract, we have deleted this sentence and now described as follow:

"The "Parade Blue" days were largely attributed to the dramatic reduction in secondary aerosols, which were also largely responsible for EPFRs and ROS reductions. The source-sector based concentrations of $PM_{2.5}$, EPFRs, G-ROS, and P-ROS during CP were reduced by 78.7%–80.8% from secondary aerosols, 59.3%–65.0% from dust sources, 65.3%–67.0% from industrial emissions, and 32.6%–43.8% from vehicle emissions, compared to the cases during NCP."

4. Line 67: this sentence should be rewritten more clearly.

**Response: Thank you for your careful review.** We apologize for not describing it clearly. We have re-written this sentence in Line 71 as the follows:

"Thus, investigating the variations in the levels and sources of ROS are vital for understanding the mechanism of ROS formation and their effect on human health."

5. Line 87: it is not clear here, the sampling information here are for PM?

**Response: Sorry for the confusion.** Indeed, all ambient samples, not just $PM_{2.5}$ samples, were collected on the rooftop of a five-floor building at the Institute of Remote Sensing and Digital Earth, Chinese Academy of Sciences. According to your comment, we have made the modifications in Line 91 and also moved the information on monitoring of gaseous pollutant to this paragraph, and details are as follows:

"All sampling were conducted on the rooftop of a five-floor building at the Institute of Remote Sensing and Digital Earth, Chinese Academy of Sciences (117.39°E, 40.01°N), which is located between the fourth and fifth ring road in northern Chaoyang District, Beijing, China and is surrounded by residential buildings and Olympic Forest Park. A total of 76 $PM_{2.5}$ samples including 38 daytime (8:00–20:00) and 38 nighttime (20:00–8:00 the next day) samples were collected on prebaked quartz filters using Digital high-flow sampler (DHA-80, Digital, Switzerland) with a flow rate of 500 L/min from August 13 to September 19, 2015. The samples were wrapped in aluminum foil and then stored in a refrigerator at -20°C until analysis. Real-time $SO_2$, $NO_2$, and $O_3$ concentrations were simultaneously monitored online by an $SO_2$ analyzer (Model 43i, Thermo Scientific, USA), NOx analyzer (Model 42i, Thermo Scientific, USA), and ozone analyzer (Model 49i, Thermo Scientific, USA), respectively."

6. Line 98: "elementary" should be "elemental".

**Response: Thanks for your kind reminder.** We have replaced "elementary" with "elemental" in Line 105 according to your suggestion.

"Organic carbon (OC) and elemental carbon (EC) in $PM_{2.5}$ were measured by a thermal/optical carbon analyzer (model RT-4, Sunset Laboratory Inc. USA)."

7. Line 104: it should be "Real-time $SO_2$, $NO_2$, and $O_3$ concentrations".

**Response: Thanks for your kind reminder.** We have revised it in Line 97 as follow:

"Real-time $SO_2$, $NO_2$, and $O_3$ concentrations were simultaneously monitored online by an $SO_2$ analyzer (Model 43i, Thermo Scientific, USA), NOx analyzer (Model 42i, Thermo Scientific, USA), and ozone analyzer (Model 49i, Thermo Scientific, USA), respectively."

8. Line 176: "spin" should be "spins"

**Response: Thank you very much for your valuable advice.** We have corrected it in Line 182 as follow:

"but much higher than that in Chongqing ($7.0 \times 10^{13}$ spins/m$^3$) in 2017–2018 (Qian et al., 2020)"

9. Figure 6: The horizontal axis should be changed to continuous time.

**Response: Thank you for your careful review.** According to your suggestion, we have revised the horizontal axis label in figure 6 to continuous time as follow:

[Figure]

Figure 6: Diurnal variations in concentrations of (a) G-ROS and (b) P-ROS during the four pollution periods.

10. Line 202-210: The authors should provide some discussions of the comparison of reduction effects in G-ROS and P-ROS during control periods.

**Response: Thank you for pointing out this problem in the manuscript.** We have added a comparison of reduction effects of G-ROS and P-GOS in Line 213-215 as follows:

"The ROS activity obtained here was expressed in nmol H$_2$O$_2$ equivalents m$^{-3}$. As shown in Figure 4, the average concentrations of G-ROS and P-ROS were 17.2 nmol

$H_2O_2/m^3$ and 13.6 nmol $H_2O_2/m^3$, respectively, during NCP, decreased to 13.8 nmol $H_2O_2/m^3$ and 7.25 nmol $H_2O_2/m^3$ during period 2, and further decreased to 10.3 nmol $H_2O_2/m^3$ and 7.02 nmol $H_2O_2/m^3$ during period 3. The concentrations of ROS during CP were comparable to those observed in urban America (Wang et al., 2011) and rural China (Zhao et al., 2023; Huang et al., 2016). It is noteworthy that the impact of the control measures on G-ROS and P-ROS was different. Compared with NCP, the percentage decrease in G-ROS during CP was 24.1%, which was lower than that in P-ROS (46.9%). This difference may be related to the complex formation and transformation mechanism of G-ROS. These results further suggested that the decrease in gaseous pollutants was lower than that in particulate pollutants."

11. Line 271-273: Why the percentage contribution from vehicle emissions increased?

**Response: Thank you very much for your question.** The percentage contribution of vehicle exhaust emissions increased during the control period. On the one hand, this is attributed to the concentration decrease from this sector was smaller than those from the other major source sectors, especially factors related to secondary aerosols. On the other hand, the sampling site in this study is located near traffic arteries and are highly influenced by the traffic source. The World Athletics Championships held at the National Stadium (known as the Bird's Nest) during period 2, which resulted in a significant increase in traffic flow near the sampling site. As a result, the percentage contribution from vehicle emissions increased significantly during control period. The corresponding revision has been provided in Line 282–285 as follows:

"The percentage contribution from vehicle emissions actually increased because the concentration decrease from this sector was smaller than those from the other major source sectors (especially the factor of secondary aerosols, as discussed below). Further, the smaller decrease in concentrations from vehicle emissions may be attributed to the World Athletics Championships held at the National Stadium (known as the Bird's Nest) during period 2, which resulted in a significant increase in traffic flow near the sampling site. As a result, the percentage contribution from vehicle emissions increased significantly during CP."

12. Conclusions: In the conclusion section, the authors should include some recommendations for future research needs on ROS and EPFR.

**Response: Thank you very much for your valuable advice.** We have added some recommendations for future research on EPFRs and ROS in the conclusions section, which are now described as follows:

"The short-term air quality control measures on hazardous substances during the 2015 China Victory Day Parade in Beijing reduced the concentrations of EPFRs by 18.3%, G-ROS by 24.1%, and P-ROS by 46.9% during CP compared to NCP. Overall, the decrease in EPFRs and ROS was smaller than those for most other measured pollutants (e.g., $PM_{2.5}$, EC, elements, and $SO_2$). Although particle matter-based air quality control measures have performed well in achieving "Parade Blue", it is difficult to simultaneously reduce the negative impacts of atmosphere on human health. Given that EPFRs and ROS exhibited a significant positive correlation ($p<0.01$) with EC, secondary inorganic ions, $NO_2$, and Cd, controlling emissions of these chemical species would benefit the reduction in EPFRs and ROS pollution. The sources of EPFRs and ROS differed between day- and nighttime. EPFRs were mainly from vehicle exhaust emissions and atmospheric oxidation processes in the daytime and vehicle exhaust emissions and fossil fuel combustion in the nighttime. Vehicle exhaust, secondary aerosols, and metals from fuel combustion processes were important sources of G-ROS and P-ROS in the daytime, while vehicle exhaust and coal combustion emissions were the major contributors of P-ROS in the nighttime. The predominant sources of $PM_{2.5}$, EPFRs, G-ROS, and P-ROS during NCP were secondary aerosols, followed by vehicle emissions, but vehicle emissions surpassed secondary aerosols to become the predominant source of these chemical species during CP. The control measures implemented during CP reduced source-sector based concentrations of $PM_{2.5}$, EPFRs, G-ROS, and P-ROS by 78.7%–80.8% from secondary aerosols, 59.3%–65.0% from dust sources, 65.3%–67.0% from industrial emissions, and 32.6%–43.8% from vehicle emissions, compared to the cases during NCP. Results from this study will benefit the development of future air quality management policies targeting EPFRs and ROS.

However, the generation and transformation processes of EPFRs and ROS involve multiple complex chemical reactions, further in-depth studies are still needed to gain a complete understanding of the formation pathways of EPFRs and ROS under different environmental conditions. For example, it is necessary in future to conduct smog chamber or flow tube experiment to simulate the photochemical reactions and oxidation processes of EPFRs and ROS in the atmosphere, as well as their interactions with different chemical species. In addition, substantial efforts are needed to develop the pretreatment and analytical methods to separate various EPFRs and gain the detailed information such as g-tensors and hyperfine splitting constants to identify the specific structures of the radicals, consequently clarifying the link between EPFRs and ROS."

13. Line 390: Please check the references and unified format.

**Response: Thank you for your careful review.** We have carefully checked the format of all references and made the modifications in References list. Some examples of modified references are shown below:

"Borrowman, C. K., Zhou, S., Burrow, T. E., and Abbatt, J. P.: Formation of environmentally persistent free radicals from the heterogeneous reaction of ozone and polycyclic aromatic compounds, Phys. Chem. Chem. Phys., 18, 205–212, http://dx.doi.org/10.1039/C5CP05606C, 2016.

Chen, Q., Sun, H., Wang, J., Shan, M., Yang, X., Deng, M., Wang, Y., and Zhang, L.: Long-life type — The dominant fraction of EPFRs in combustion sources and ambient fine particles in Xi'an, Atmos. Environ., 219, 117059, http://dx.doi.org/10.1016/j.atmosenv.2019.117059, 2019c.

Guo, X., Zhang, N., Hu, X., Huang, Y., Ding, Z., Chen, Y., and Lian, H.: Characteristics and potential inhalation exposure risks of $PM_{2.5}$–bound environmental persistent free radicals in Nanjing, a mega–city in China, Atmos. Environ., 224, 117355, http://dx.doi.org/10.1016/j.atmosenv.2020.117355, 2020."

---

## Author Comment (AC2)

Dear Editor-in-Chief and Reviewers,

We are submitting a revised version of the manuscript (No.: egusphere-2023-2703), entitled: "**Measurement report: Impact of emission control measures on environmental persistent free radicals and reactive oxygen species – A short-term case study in Beijing**". We have carefully addressed all the comments provided by the reviewer, and an item-by-item response to the comments of the reviewers is given below. All revisions are highlighted in blue in the main text of the revised manuscript.

Thank you for taking care of the review process for this paper.

Sincerely,

Prof. Jihua Tan and coauthors

College of Resources and Environment, University of Chinese Academy of Sciences, Beijing 100049

tanjh@ucas.ac.cn

This measurement report reports measurements of EPFRs and ROS in Beijing, highlighting the impact of emission control and reductions in 2015. There is good evidence for their conclusions with scientific data, however their results fail to give error bars or uncertainty values that would make their conclusions stronger. I have several major comments that need to be addressed as below.

Major comments:

1. I found that introduction misses some key information and references are missing. I suggest citing several pioneering works on EPFR by Dellinger and coworkers, by making it clear that EPFRs are mainly generated by incomplete combustion and traffic emissions. In addition, a recent study has shown that biomass burning is a significant source of EPFRs (Fang et al., DOI: 10.1039/d2ea00170e). I would also mention in introduction that heterogeneous chemistry could be a source of EPFRs (Borrowman et al., DOI: 10.1039/C5CP05606C).

**Response: We are appreciative of the reviewer's suggestion.** We have supplemented the key information on the sources of EPFRs and added these reference to the reference list according to your suggestion:

Line 58-70: "EPFRs are primarily derived from all most incomplete combustion sources such as vehicle exhaust, biomass burning, and coal combustion (Wang et al., 2019b; Dugas et al., 2016; Saravia et al., 2013). EPFRs can be formed and stabilized on the surface of particulate matter containing transition metals and substituted aromatic structures emitted during combustion processes (Odinga et al., 2020; Chen et al., 2019a). For example, The incomplete combustion of vehicle emissions has been identified as an important source of EPFRs (Chen et al., 2018b). Dellinger et al. (2001) have shown that EPFRs in $PM_{2.5}$ are associated with combustion sources. Fang et al. (2023) found that high concentrations of EPFRs are emitted from biomass burning. In addition to the combustion sources, EPFRs can also result from secondary processes in the atmosphere. It has been reported that EPFRs can be formed by the heterogeneous reaction of $O_3$ and polycyclic aromatic compounds (Borrowman et al., 2016a). EPFRs can also be formed from polycyclic aromatic hydrocarbons (PAHs) after photolysis (Li et al., 2022). Moreover, a recent study shows that EPFRs may also derive from dust sources (Li et al., 2023). Chen et al. (2018a) found that dust storms can increase the concentration of EPFRs in $PM_{2.5}$, and metal oxides contained within dust particles provide the prerequisite conditions for EPFRs formation."

2. Secondary organic aerosols (SOA) have been shown to be a significant source of ROS (Venkatachari et al., DOI: 10.1080/02786820601116004; Wang et al., DOI: 10.1080/02786826.2011.633582), including OH, superoxide, H2O2 and organic radicals (Tong et al., DOI: 10.1021/acs.est.8b03695, Wei et al., DOI: 10.1021/acs.est.0c07789). This should be mentioned/discussed especially because secondary processes are found to be source of ROS in this study.

**Reply: Thank you very much for your valuable advice.** The corresponding revision has been provided in the revised manuscript as follows:

Line 75-83: "Multiple sources of ROS have been identified, including wood combustion (Zhou et al., 2018), vehicle exhaust (Verma et al., 2010), and cooking emissions (Wang et al., 2020a). In addition, many studies have demonstrated that secondary sources related to photochemical reactions and oxidation reactions may be an important source of ROS. Volatile organic compounds (VOCs) and NOx have been shown to produce ROS through photochemical reactions (Venkatachari et al., 2007). ROS can also form on the surface of particles or in air through reactions with ozone ($O_3$) under dark conditions (Zhu et al., 2018). $H_2O_2$ in secondary organic aerosols (SOA) has been found to be as much or more than ambient particles (Wang et al., 2012). OH$^{\bullet}$ and organic radicals can be formed from isoprene and β-pinene SOA, whereas $H_2O_2$ and $O_2^{\bullet-}$ mainly from naphthalene SOA has also been proposed (Tong et al., 2018a; Wei et al., 2021)."

3. As using Mg2+ and Ca2+ standards to calibrate EPFR for g-factor and absolute spin amount is uncommon, please elaborate on this procedure.

**Reply: Sorry for our mistake.** For wide magnetic field scans, linear offset corrections are required and a sample with an EPR spectra with large splitting are required for a calibration sample. $Mg^{2+}$ and $Cr^{3+}$ can be used for this calibration. Both of these standards have been proven effective for calibrating the g-factor and absolute spin number of EPFRs (Chen et al, 2019). During the calibration process, the $Mg^{2+}$ and $Cr^{3+}$ standard samples were inserted into the resonator and the system was tuned. The spectrometer was used to perform EPR signal scans on the $Mg^{2+}$ and $Cr^{3+}$ standards, and then the field offset was set to zero to ensure that the signal measured by the instrument matches exactly the signal for $Mg^{2+}$ and $Cr^{3+}$. According your suggestion, we have added the procedure for calibrating the g-factor and absolute spin number of EPFRs using metal standard substances in Line 135-139. The modification is as follows:

"The absolute spin amount and g factor were calibrated with $Mg^{2+}$ and $Cr^{3+}$ standards. Both of these standards have been proven effective for calibrating the g-factor and absolute spin number of EPFRs (Chen et al., 2019a; Chen et al., 2019b). During the calibration process, the $Mg^{2+}$ and $Cr^{3+}$ standard samples were inserted into the resonator and the system was tuned. The field offset was set to zero to ensure that the signal measured by the instrument matches exactly the signal for $Mg^{2+}$ and $Cr^{3+}$."

Reference:

Chen Q, Sun H, Wang M, et al. Environmentally persistent free radical (EPFR) formation by visible-light illumination of the organic matter in atmospheric particles[J]. Environmental Science and Technology, 2019, 53(17): 10053-10061.

4. For measurements of ROS, are they mostly H2O2? Could your measurements also sensitive to other short-lived ROS such as OH and superoxide, or organic hydroperoxides? Please make it clear in the method section, as it is ambiguous throughout the manuscript.

**Response: We are grateful for the suggestion.** We used the DCFH method to determine the concentrations of G-ROS and P-ROS. DCFH method has the lowest specificity and selectivity for different ROS species, and is capable of reacting with multiple ROS, including $H_2O_2$, as well as other short-lived ROS, such as OH radicals, superoxide radicals, peroxyl radicals, and peroxynitrite (Bates et al, 2019). Following your suggestion, we have made the modification in the Methods and Materials section as follows:

Line 154-158: "The DCFH method has the lowest specificity and selectivity for different types of ROS, capable of reacting with multiple ROS, including $H_2O_2$, as well as other short-lived ROS, such as OH radicals, superoxide radicals, peroxyl radicals, and peroxynitrite (Bates et al., 2019)."

Reference:

Bates J T, Fang T, Verma V, et al. Review of acellular assays of ambient particulate matter oxidative potential: methods and relationships with composition, sources, and health effects[J]. Environmental science and technology, 2019, 53(8): 4003-4019.

5. Semi-quinone radicals are regarded as O-centered radicals (L116), but they have resonance structure and can have an unpaired electron on a carbon atom.

**Response: Thank you for your valuable comments.** We have revised it in Line 143-144 according to your suggestion as follows:

"while EPFRs with g-factor of 2.004 and above are designated as oxygen-centered free radicals, such as semiquinone radicals (Zhu et al., 2019). Notably, semiquinone radicals have a resonance structure and can have an unpaired electron on the carbon atom."

6. It was interesting to see a positive correlation of O3 with EPFRs, given that a previous study has observed a negative correlation (Hwang et al., DOI: 10.1021/acsearthspacechem.1c00135). Please discuss the difference.

**Response: Thanks for your kind reminder.** In our study, we observed that daytime EPFRs were positively correlated with $O_3$, suggesting that some of the EPFRs may originate from secondary processes associated with $O_3$, which is consistent with previous studies (Chen et al, 2019; Borrowman et al, 2016). However, contrary to the results of Hwang et al. (Hwang et al, 2021), they found negative correlation between EPFRs and $O_3$ at the highway site and believed that it might be due to the titration of $O_3$ by NO resulting in a negative correlation between $O_3$ and NO. This difference may be due to different study regions and environmental conditions Our study was conducted during the intense sunlight of summer, favorable for $O_3$ formation, where high $O_3$ concentrations may promote the formation of EPFRs in particulate matter (Borrowman et al, 2016). For example, EPFRs typically form from the heterogeneous reaction between polycyclic aromatic hydrocarbons and $O_3$. However, at highway sites, pollutants such as NOx from vehicle emissions may have different effects on air chemistry, resulting in a negative correlation between EPFRs and $O_3$. According to your suggestion, we have carefully revised it in Line 290-296 and now described as:

"Meanwhile, a significant positive correlation between ERFRs and $O_3$ was also observed in the daytime ($p<0.1$), consistent with the results of Chen et al (2019b). The oxidation of different types of PAHs by $O_3$ could form different types of EPFRs, as demonstrated in a previous study (Borrowman et al., 2016b). However, Huang et al. observed a negative correlation between EPFRs and $O_3$ at highway sites and believed that this may be due to the consumption of $O_3$ by NO (Hwang et al., 2021). In this study, the hot summer conditions were more conducive to the conversion of PAHs into EPFRs, especially in the presence of high $O_3$ concentrations. This implies that the mechanism of EPFRs generation varies under different environmental conditions."

Chen Q, Sun H, Mu Z, et al. Characteristics of environmentally persistent free radicals in $PM_{2.5}$: Concentrations, species and sources in Xi'an, Northwestern China[J]. Environmental pollution, 2019, 247: 18-26.

Borrowman C K, Zhou S, Burrow T E, et al. Formation of environmentally persistent free radicals from the heterogeneous reaction of ozone and polycyclic aromatic compounds[J]. Physical Chemistry Chemical Physics, 2016, 18(1): 205-212.

Hwang B, Fang T, Pham R, et al. Environmentally persistent free radicals, reactive oxygen species generation, and oxidative potential of highway PM$_{2.5}$[J]. ACS Earth and Space Chemistry, 2021, 5(8): 1865-1875.

7. I found Sect. 3.5 not robust or justified well. First, please discuss each factor more in detail and why each factor is assigned to have specific source. Such discussion is completely missing, so it is totally unclear why top factor is assigned to be secondary aerosols, 2nd as vehicle, etc. What are key tracers and features in each factor? This needs to be elaborated. I also do not understand fully, how Fig. 9 is constructed from Fig.

**Response: Thanks for your kind reminder.** We have added the reasons for each factor to be identified as specific sources in Line 323-327 of the manuscript according to your suggestions. Additionally, we have added a description of how the factor contributions in Figure 9 were obtained from the PMF model in the Methods and Materials section.

Line 323-327: "The high proportion of NO$_3^-$, SO$_4^{2-}$, and NH$_4^+$ are attributed to secondary aerosols. A factor is recognized as vehicle emissions due to the high abundance of EC and Cu. Another factor can be recognized as dust sources because of the high proportion of Mg, Al, Ca, and Fe. A fourth factor was linked to industrial emissions sources due to the high proportion of V, Mn, Rb, Cd, Pb, and Bi. Additionally, another factor was identified as other sources because of the high abundance of Co and Zn."

Line 162-169: "The fundamental principle of PMF involves first calculating the errors of various chemical components in particulate matter using weights, followed by utilizing the least squares method to estimate the main pollution sources of the particulate matter and their contribution. The PMF model decomposes a matrix of specific sample data (X) into a source contributions matrix (G) and factor profile matrix (F), as well as a residual matrix (E), as shown in the following equation:

$$X_{ij} = \sum_{k=1}^{p} g_{ik} f_{kj} + e_{ij} \tag{1}$$

Where $X_{ij}$ denotes the concentration of the $j$th species in the $i$th sample, $g_{ik}$ represents the source contribution of the $k$th factor to the $i$th sample, $f_{kj}$ is the factor profile of $j$th species in the $k$th factor, and $e_{ij}$ is the residual matrix."

8. How did you quantify contributions of each factor (or source) to EPFR or ROS, even though the strength or contribution of EPFR or ROS in each factor is different? Without these clarifications, the results of Fig. 9 is highly questionable and would suggest omitting this analysis if not clarified.

**Response: Thank you for your valuable suggestions.** In the process of using the PMF model for analysis, we are aware that this model, as a mathematical tool, transcends single applications and includes, but is not limited to, atmospheric particulate matter source apportionment. Regarding the issue you raised about the inconsistency in units between EPFRs/ROS and $PM_{2.5}$ component concentrations, it is important to note that the decomposition process of the PMF model focuses on the relative relationships among various components. Therefore, even if the units of EPFRs/ROS differ from those of other components, the model can still assess their relative contributions within the component matrix, with emphasis on their relative proportions within that matrix. Moreover, we have noted that researchers have successfully combined non-traditional parameters such as EPFRs/ROS with the PMF model for source apportionment (Ainur et al, 2023; Wang et al, 2019). We also mention this in Line 160-161 of the manuscript. To enhance transparency and clarity, we have clearly labeled the concentration units of EPFRs and ROS in figures 8. We appreciate the opportunity you have provided and look forward to your further feedback.

Line 160-161: "Researchers have successfully employed PMF for source apportionment o EPFRs and ROS (Ainur et al., 2023; Wang et al., 2019b). In this study, we used the Environmental Protection Agency (EPA) PMF 5.0 version to perform the source apportionment of $PM_{2.5}$, EPFRs, G-ROS, and P-ROS."

[Figure]

Reference:

Ainur D, Chen Q, Sha T, et al. Outdoor health risk of atmospheric particulate matter at night in Xi'an, Northwestern China[J]. Environmental Science and Technology, 2023, 57(25): 9252-9265.

Wang Y, Li S, Wang M, et al. Source apportionment of environmentally persistent free radicals (EPFRs) in PM$_{2.5}$ over Xi'an, China[J]. Science of the total environment, 2019, 689: 193-202.

9. Error analysis and uncertainties are missing for some figures. Please include error bars in Fig. 1, 2, 4, 5, and possibly also 6.

**Response: Thank you for your suggestion to include error bars in Figures 1, 2, 4, 5, and possibly 6.** The online monitoring techniques used in the study usually do not involve the measurement of parallel samples, which is common practice for this type of analysis. Therefore, we are unable to provide error bars for each data monitored in real time. However, after your reminder, to increase the transparency and credibility of the data, we have added the standard deviation to the mean data described in the text,

as modified below:

Line 215: "The average concentration of EPFRs was $(1.00\pm0.75)\times10^{14}$ spins/m$^3$ during

NCP and $(8.19\pm5.60)\times10^{13}$ spins/m$^3$ during CP,"

Line 255-258: "the average concentrations of G-ROS and P-ROS were $17.2\pm2.51$ nmol H$_2$O$_2$/m$^3$ and $13.6\pm2.71$ nmol H$_2$O$_2$/m$^3$, respectively, during NCP, decreased to $13.8\pm1.29$ nmol H$_2$O$_2$/m$^3$ and $7.25\pm1.79$ nmol H$_2$O$_2$/m$^3$ during period 2, and further decreased to $10.3\pm0.63$ nmol H$_2$O$_2$/m$^3$ and $7.02\pm0.57$ nmol H$_2$O$_2$/m$^3$ during period 3."

10. Line 171: Please include the standard deviation for this reduction 18.3 ± ??.? %. Are three significant figures appropriate considering error calculations? It appears that the EPFRs are actually highest in the strict control period. Comment on this.

**Response: Thank you for the above suggestion.**

We acknowledge that the online monitoring techniques utilized in our study do not include the measurement of parallel samples, which means we cannot calculate the standard deviation for the 18.3% reduction in EPFRs during the control period compared to the non-control period. However, after your reminder, we have added the standard deviation to the average value described in the manuscript at Lines 215 and 255-258. Additionally, the average concentration of EPFRs was $1.00\times10^{14}$ spins/m$^3$ during non-control period and $8.19\times10^{13}$ spins/m$^3$ during control period. It is observable that there was a decrease in EPFRs concentration during control period compared to the non-control period. Notably, the EPFRs concentration increased during period 3. Although the emission intensity from pollution sources has decreased under strict control measures, the influence of certain characteristic sources on EPFRs formation may still be relatively minor. Chen et al.(2020) showed that the change in EPFRs concentrations is unrelated to the change in the PM concentration, but rather determined by their source characteristics. For instance, traffic emissions are a significant source of EPFRs in PM, and it is speculated that activities during the parade may have been influenced by traffic sources, thus increasing EPFRs concentration. As evident from the factor analysis section, there was an observed increase in contributions from traffic sources during the control period. According to your suggestions, we have made the modification in the revised manuscript as follows:

Line 215: "The average concentration of EPFRs was $(1.00\pm0.75)\times10^{14}$ spins/m$^3$ during

NCP and $(8.19\pm5.60)\times10^{13}$ spins/m$^3$ during CP,"

Line 255-258: "the average concentrations of G-ROS and P-ROS were 17.2±2.51 nmol $H_2O_2/m^3$ and 13.6±2.71 nmol $H_2O_2/m^3$, respectively, during NCP, decreased to 13.8±1.29 nmol $H_2O_2/m^3$ and 7.25±1.79 nmol $H_2O_2/m^3$ during period 2, and further decreased to 10.3±0.63 nmol $H_2O_2/m^3$ and 7.02±0.57 nmol $H_2O_2/m^3$ during period 3."

Line 214-222: "The average concentration of EPFRs was $(1.00±0.75)×10^{14}$ spins/$m^3$ during NCP and $(8.19±5.60)×10^{13}$ spins/$m^3$ during CP, which represents 18.1% lower concentration during CP than NCP. The percentage decrease in EPFRs was smaller than most of the other measured pollutants ($PM_{2.5}$, EC, elements, $NO_2$, and $SO_2$). Notably, the concentration of EPFRs increased during period 3, despite the reduction in emission intensity from pollution sources under strict control measure conditions, suggesting the impact on the formation of EPFRs from certain characteristic sources still be modest. Chen et al.(2020) showed that the change in the EPFRs concentrations is unrelated to the change in the PM concentration, but rather determined by their source characteristics. For instance, activities during the parade may have increased the contribution from traffic and other sources. Detailed discussion on these source characteristics will follow in subsequent source apportionment sections"
Reference:
Chen Q, Sun H, Song W, et al. Size-resolved exposure risk of persistent free radicals (PFRs) in atmospheric aerosols and their potential sources[J]. Atmospheric Chemistry and Physics, 2020, 20(22): 14407-14417.

11. Line 270: Before it was mentioned that more complex formation of EPFRs happened during control periods, possibly from secondary reactions. However, based on the PMF SOA decreased, as well as precursor gasses of secondary aerosols, and vehicles were the largest source. Please comment on whether these two conclusions agree or disagree with each other. There is an apparent increase in "other sources" that could tie into this complex formation of EPFRs during the CP.

**Response: Thank you for your careful review.** The issues you mention are very important. These two conclusions are to some extent interrelated. First, the complex formation of EPFRs during control period may be associated with secondary reactions, leading to the generation of substances that could trigger a more intricate EPFRs formation process. Secondly, PMF analysis indicates that the decrease in SOA and its precursor gases may also influence the formation of EPFRs, resulting in a reduction in the production of more complex EPFRs. However, the formation of EPFRs is a complex

process that may involve multiple sources and intricate atmospheric chemistry. You are quite right that we did not consider the complexity of EPFR formation, which could also be attributed to increased contributions from "other sources", such as uncontrolled natural sources and regional transport, potentially leading to a more intricate formation of EPFRs during control period. According to your suggestions, we have revised it in Line 246-251:

"The average $\Delta H_{p\text{-}p}$ of EPFRs during CP and NCP was 4.62 ± 1.06 G and 4.42 ± 0.87 G, respectively. The slightly larger $\Delta H_{p\text{-}p}$ during CP than NCP indicates a relatively complex path for the formation of EPFRs under strict control measure conditions. This may be explained by a marked increase in the activity of other sources, which will be discussed below. However, due to the complex formation and transformation of EPFRs, current evidence does not sufficiently explain the changes in EPFRs, necessitating further investigation to uncover deeper mechanisms."

Other minor comments:

1. Line 34: And/Or used in this context is unclear, consider rephrasing.

**Response: We are grateful for the suggestion.** We have replaced all "and/or" with "or" in the revised manuscript as follows:

Line 32: "The contribution of G-ROS to the atmospheric oxidizing capacity increased or that of P-ROS decreased during CP compared to NCP."

Line 264: "Furthermore, the much higher ratios of G-ROS to P-ROS during CP than NCP suggested that the contribution of G-ROS to atmospheric oxidizing capacity was increased or that of P-ROS was decreased during this period (Figure 5)."

2. Line 41: Grammatical correction- should be changed to "enables free radicals to be highly reactive"

**Response: We are very sorry for our careless mistake.** Based on the revisions, I've revised the abstract correspondingly and removed this sentence, revising it to the following:

"EPFRs in the non-control period (NCP) tended to be radicals centered on a mixture of carbon and oxygen, while those in the control period (CP) were mainly oxygen-centered free radicals. The contribution of G-ROS to the atmospheric oxidizing capacity increased or that of P-ROS decreased during CP compared to NCP. The strict control measures reduced ambient EPFRs, G-ROS, and P-ROS by 18.3%, 24.1%, and 46.9%, respectively, which were smaller than the decreases of most other measured pollutants.

Although particle matter-based air quality control measures have performed well in achieving "Parade Blue", it is difficult to simultaneously reduce the negative impacts of atmosphere on human health. The "Parade Blue" days were largely attributed to the dramatic reduction in secondary aerosols, which were also largely responsible for EPFRs and ROS reductions. The source-sector based concentrations of $PM_{2.5}$, EPFRs, G-ROS, and P-ROS during CP were reduced by 78.7%–80.8% from secondary aerosols, 59.3%–65.0% from dust sources, 65.3%–67.0% from industrial emissions, and 32.6%–43.8% from vehicle emissions, compared to the cases during NCP. Furthermore, vehicle emissions and other inadequately controlled pollution sources may play a more complex role than expected in air quality and public health. This insight will prompt policymakers to reevaluate current air quality management strategies to more effectively address the challenges posed by pollutants such as EPFRs and ROS."

3. Line 51: Consider including your source of dust formation of EPFRs to this sentence in addition to the following one.

**Response: Thank you very much for your valuable advice.** According to your suggestion, I have added a discussion of the sources of dust formation in EPFRs in the manuscript as follows:

Line 68-70: "Moreover, a recent study shows that EPFRs may also derive from dust sources (Li et al., 2023). Chen et al. (2018a) found that dust storms can increase the concentration of EPFRs in $PM_{2.5}$, and metal oxides contained within dust particles provide the prerequisite conditions for EPFRs formation."

4. Line 56: Not the correct source- shouldn't it be Gehling 2014 (Environ. Sci. Technol. 2014, 48, 8, 4266–4272)?

**Response: Thanks for your kind reminder.** We have revised it and cited related references in Line 70-71 as follows:

"EPFRs have received widespread attention in recent years because of their ability to convert $O_2$ molecules into reactive oxygen species (ROS) (Gehling et al., 2014)."

5. Line 74: Detail the short-term emission measures put in place that may be relevant to this study.

**Response: We are appreciative of the reviewer's suggestion.** We have provided detailed descriptions of the short-term emission control measures implemented during the study period in Table S2 of the Supplementary Material. As shown in the table below:

**Table S2: Descriptions of control measures implemented in different periods. "√" denotes the control.**

| | Control measures | Period 1 | Period 2 | Period 3 | Period 4 |
|---|---|---|---|---|---|
| Long-term routine measures | | √ | √ | √ | √ |
| | Odd-and-even plate rule for vehicle use | | √ | √ | |
| | Polluting industry restriction | | √ | √ | |
| | Construction sites shutdown | | √ | √ | |
| | Delay of school opening | | √ | √ | |
| Short-term routing measures | Vacation days off | | | √ | |
| | More frequent road sprinkling | | √ | √ | |
| | Road traffic control around Tiananmen Square | | | √ | |
| | Museum and tourist attraction closedowns | | | √ | |
| | Enhanced measures in surrounding cities | | | √ | |

6. Line 76: List other pollutants that have been measured and their importance to your study.

**Response: Thank you very much for your valuable advice.** Zhao et al. (2017) and Wang et al. (2017) showed that the concentrations of primary organic aerosols (POA), SOA, water-soluble ions, and gaseous pollutants decreased significantly during control period, highlighting the effectiveness of short-term control measures. However, the potential impacts of these measures on public health, especially regarding EPFRs and ROS, remains unclear and require further investigation. According to your comments, we have included a list of the pollutants measured in these studies, along with a discussion of their importance to our study. Here are the details:

Line 95-100: "Particle concentrations in Beijing were substantially reduced during this period, achieving the so-called "Parade Blue" (Huang et al., 2018b). Other air pollutants, such as primary organic aerosols (POA), SOA, water-soluble ions, and gaseous pollutants also exhibited significant reductions during this period (Zhao et al., 2017; Wang et al., 2017), demonstrating the potential of short-term control measures in reducing air pollution. However, the potential impacts of these measures on public health, especially regarding EPFRs and ROS, remain unclear. This event also provided an excellent opportunity to quantify the effectiveness of control measures on EPFRs and ROS."

7. Line 96: What are the differences between "regularly control measures" and "stricter control measures"? Detail the control measures put in place.

**Response: Thank you very much for your advice.** We have described the control measures in Table S2 of the Supplementary Material, where we outlined the specific

control measures taken during different periods. Period 2 having regularly control measures, including odd-and-even plate rule for vehicle use, polluting industry restriction, construction sites shutdown, delay of school opening, and more frequent road sprinkling. In response to China Victory Day Parade, the stricter control measures were implemented during period 3 to achieve more significant emission reductions. These measures included odd-and-even plate rule for vehicle use, polluting industry restriction, construction sites shutdown, delay of school opening, vacation days off, more frequent road sprinkling, road traffic control around Tiananmen Square, museum and tourist attraction closedowns, and enhanced measures in surrounding cities. In order to get a clear understanding of the control measures used at different periods, we have modified it in Line 119 of the revised manuscript as follows:

"The whole sampling period is divided into four sub-periods for analysis, with the specific control measures for each sub-period presented in Table S2."

**Table S2: Descriptions of control measures implemented in different periods. "√" denotes the control.**

| | Control measures | Period 1 | Period 2 | Period 3 | Period 4 |
|---|---|---|---|---|---|
| Long-term routine measures | | √ | √ | √ | √ |
| | Odd-and-even plate rule for vehicle use | | √ | √ | |
| | Polluting industry restriction | | √ | √ | |
| | Construction sites shutdown | | √ | √ | |
| | Delay of school opening | | √ | √ | |
| Short-term routing measures | Vacation days off | | | √ | |
| | More frequent road sprinkling | | √ | √ | |
| | Road traffic control around Tiananmen Square | | | √ | |
| | Museum and tourist attraction closedowns | | | √ | |
| | Enhanced measures in surrounding cities | | | √ | |

8. Line 96: May be clearer to say either "regulatory control measures" or "regular control measures."

**Response: Thank you for your suggestion.** We have already defined the control measures implemented during period 2 as "regularly control measures". To clearly identify the stricter control measures implemented during period 3, and distinguish it from the regularly control measures taken during period 2, we have chosen to retain the term "stricter control measures". We appreciate your attention to terminology clarification.

9. Line 120: Define GAC-ROS

**Response: Thank you for your careful review.** We have defined GAC-ROS in Line

148 of the revised manuscript as follows:

"The gas and aerosol collector-ROS (GAC-ROS) online monitoring system was used to measure the concentrations of G-ROS and P-ROS."

10. Line 127: Specify what was used as standards. "Flesh standards" should be changed to "fresh standards."

**Response: We are very sorry for our careless mistake.** We have revised it in Line 157-158 according to your suggestions, and details are as follows:

"For data accuracy, fresh DCFH and HRP were prepared at least every two days, and $H_2O_2$ standard curves were created daily."

11. Line 143: Specify which elements are measured, is this from ICP-MS?

**Response: Thank you for the above suggestion.** For elemental measurements, including Li, Na, Mg, Al, K, Ca, V, Mn, Fe, Co, Cu, Zn, As, Se, Rb, Cd, Pb, and Bi was determined using ICP-MS. Indeed, the method for determining these elements is already described in the Methods and Materials section as follow:

Line 126-129: "Elements (Li, Na, Mg, Al, K, Ca, V, Mn, Fe, Co, Cu, Zn, As, Se, Rb, Cd, Pb, and Bi) in $PM_{2.5}$ were extracted by microwave digestion with 7 mL of ultrapure water, 2 mL of $HNO_3$, and 1 mL of $H_2O_2$, and the concentrations of elements were detected using inductively coupled plasma-mass spectrometry (ICP-MS)."

12. Line 148: What is MMW-PAHs?

**Response: Thank you for your careful review.** The MMW-PAHs here refers to the middle molar weight PAHs. We explained MMW-PAHs in Line 182 as follow:

"temporal variations of $PM_{2.5}$, EC, middle molar weight PAHs (MMW-PAHs, 4 ring PAHs), elements, and gas pollutants were first examined."

13. Line 151: There are 3 percentages listed for decreases by NO2 and SO2, but there should only be 2. Remove the percentage not associated with NO2 and SO2 decrease. The first percentage may refer to O3, in which O3 should be added to the sentence.

**Response: We are very sorry for our careless mistake.** You are right that the "$O_3$" was missed here, and we have modified it in Line 190 as follows:

"Regarding the gaseous pollutants, the concentrations of $O_3$, $SO_2$, and $NO_2$ decreased by 10.8%, 51.2%, and 45.5%, respectively,"

14. Line 154: State the decrease in O3 concentration.

**Response: We are very sorry for our careless mistake.** We have added a description of the reduction in $O_3$ concentration during control period in Line 190 as follows:

"Regarding the gaseous pollutants, the concentrations of $O_3$, $SO_2$, and $NO_2$ decreased by 10.8%, 51.2%, and 45.5%, respectively, during CP compared to those in the NCP."

15. Line 159: Elaborate on why there would be increased traffic at night. Is this common for this location?

**Response: We appreciate the reviewer's observation regarding the increased traffic at night.** Our analysis indicates that this phenomenon can be attributed to several factors. Firstly, the restriction of heavy-duty vehicle to enter Beijing only at night may contribute to increased emissions from diesel vehicles near the fourth and fifth ring roads at nighttime (Cai et al., 2020). Secondly, lower temperatures and reduced solar radiation at night decrease the photolysis of $NO_2$, which is the main chemical mechanism for $NO_2$ loss during the day (Cai and Xie, 2010), further leading to elevated $NO_2$ concentrations at nighttime. Similar increases in nighttime traffic have been observed in other studies conducted in Beijing (Lin et al., 2009; Ke et al., 2017; Cai et al., 2020). Thanks to your reminder, to explain in more detail the reasons for increased nighttime traffic and to explore the universality of this phenomenon in Beijing, we have revised it in Line 198-206 in the manuscript as follow:

"The average concentrations of EC and $NO_2$ were generally higher during the nighttime (1.33 $\mu g/m^3$ and 40.2 $\mu g/m^3$, respectively) than daytime (0.82 $\mu g/m^3$ and 28.4 $\mu g/m^3$, respectively) in the whole measurement period. This is especially the case during the NCP, likely due to increased nighttime traffic emissions or the occurrence of temperature inversions (Yang et al., 2015; Wu et al., 2012). During daytime, the restrictions on heavy-duty vehicles entering the urban areas of Beijing may lead to increased emissions from diesel vehicles near the fourth and fifth ring roads at nighttime (Cai et al., 2020). Similar diurnal variations have also been observed previously in Agra and Beijing (Pipal et al., 2014; Lin et al., 2009; Ke et al., 2017; Cai et al., 2020). Additionally, lower temperatures and reduced solar radiation at nighttime decrease the photolysis of $NO_2$, which is the main chemical mechanism for $NO_2$ loss at daytime (Cai and Xie, 2010), further contributing to the elevated $NO_2$ concentrations at nighttime."

Reference:

Cai, J., Chu, B., Yao, L., Yan, C., Heikkinen, L. M., Zheng, F., Li, C., Fan, X., Zhang, S., Yang, D., Wang, Y., Kokkonen, T. V., Chan, T., Zhou, Y., Dada, L., Liu, Y., He, H.,

Paasonen, P., Kujansuu, J. T., Petäjä, T., Mohr, C., Kangasluoma, J., Bianchi, F., Sun, Y., Croteau, P. L., Worsnop, D. R., Kerminen, V. M., Du, W., Kulmala, M., and Daellenbach, K. R.: Size-segregated particle number and mass concentrations from different emission sources in urban Beijing, Atmos. Chem. Phys., 20, 12721–12740.

Lin, P., Hu, M., Deng, Z., Slanina, J., Han, S., Kondo, Y., Takegawa, N., Miyazaki, Y., Zhao, Y., and Sugimoto, N.: Seasonal and diurnal variations of organic carbon in PM2.5 in Beijing and the estimation of secondary organic carbon, J. Geophys. Res. Atmos., 114.

Ke, W., Zhang, S., Wu, Y., Zhao, B., Wang, S., and Hao, J.: Assessing the Future Vehicle Fleet Electrification: The Impacts on Regional and Urban Air Quality, Environ. Sci. Technol., 51, 1007–1016.

16. Line 180: EPFRs are also formed during irradiation. Please elaborate on what EPFRs may be converting to during irradiation, or include a source for this theory.

**Response: Thank you very much for your suggestion.** Previous studies have shown that the half-life times of EPFRs were shorter under light than dark conditions (Lang et al., 2022; Chen et al., 2019), suggesting that light irradiation promotes the transformation of EPFRs (Jia et al., 2019). For instance, semiquinone radicals can rapidly degrade into $CO_2$ under light irradiation conditions (Li et al., 2014). According to your suggestions, we have revised it in Line 227-231 of the revised manuscript as follow:

"The lower EPFRs concentrations during daytime may be related to the rapid conversion of EPFRs to other chemical species under strong irradiation. Previous studies have shown that the half-life times of EPFRs were shorter under light than dark conditions (Lang et al., 2022; Chen et al., 2019a), suggesting that light irradiation promotes the transformation of EPFRs (Jia et al., 2019). For instance, semiquinone radicals can rapidly degrade into $CO_2$ under light irradiation conditions (Li et al., 2014)."

Reference:

Lang, D., Jiang, F., Gao, X., Yi, P., Liu, Y., Li, H., Chen, Q., Pan, B., and Xing, B.: Generation of environmentally persistent free radicals on faceted TiO2 in an ambient environment: roles of crystalline surface structures, Environ. Sci. Nano., 9, 2521–2533, 2022.

Chen, Q., Sun, H., Wang, M., Wang, Y., Zhang, L., and Han, Y.: Environmentally persistent free radical (EPFR) formation by visible-light illumination of the organic matter in atmospheric particles, Environ. Sci. Technol., 53, 10053–10061, 2019.

Jia, H., Zhao, S., Shi, Y., Zhu, K., Gao, P., and Zhu, L.: Mechanisms for light-driven evolution of environmentally persistent free radicals and photolytic degradation of PAHs on Fe(III)-montmorillonite surface, J. Hazard. Mater., 362, 92–98, 2019.

Li, H., Pan, B., Liao, S., Zhang, D., and Xing, B.: Formation of environmentally persistent free radicals as the mechanism for reduced catechol degradation on hematite-silica surface under UV irradiation, Environ. Pollut., 188, 153–158, 2014.

17. Line 181: Similar to my comment above, please include a source for why it is thought that traffic increased at night, as it is unlikely in other areas.

**Response: We appreciate the reviewer's observation regarding the increased traffic at night.** Our analysis indicates that this phenomenon can be attributed to several factors. Firstly, the restriction of heavy-duty vehicle to enter Beijing only at night may contribute to increased emissions from diesel vehicles near the fourth and fifth ring roads at nighttime (Cai et al., 2020). Secondly, lower temperatures and reduced solar radiation at night decrease the photolysis of $NO_2$, which is the main chemical mechanism for $NO_2$ loss during the day (Cai and Xie, 2010), further leading to elevated $NO_2$ concentrations at nighttime. Similar increases in nighttime traffic have been observed in other studies conducted in Beijing (Lin et al., 2009; Ke et al., 2017; Cai et al., 2020). Thanks to your reminder, we have revised it in Line 197-205 in the manuscript as follow:

"The average concentrations of EC and $NO_2$ were generally higher during the nighttime (1.33 μg/m$^3$ and 40.2 μg/m$^3$, respectively) than daytime (0.82 μg/m$^3$ and 28.4 μg/m$^3$, respectively) in the whole measurement period. This is especially the case during the NCP, likely due to increased nighttime traffic emissions or the occurrence of temperature inversions (Yang et al., 2015; Wu et al., 2012). During daytime, the restrictions on heavy-duty vehicles entering the urban areas of Beijing may lead to increased emissions from diesel vehicles near the fourth and fifth ring roads at nighttime (Cai et al., 2020). Similar diurnal variations have also been observed previously in Agra and Beijing (Pipal et al., 2014; Lin et al., 2009; Ke et al., 2017; Cai et al., 2020). Additionally, lower temperatures and reduced solar radiation at nighttime decrease the photolysis of $NO_2$, which is the main chemical mechanism for $NO_2$ loss at daytime (Cai and Xie, 2010), further contributing to the elevated $NO_2$ concentrations at nighttime."

18. Figure 2/Line 185: Include error bars in this graph if available.

**Response: Thank you for your suggestion above.** Regarding the error bars in Figure

2, we would like to clarify that the online monitoring techniques used in the study usually do not involve the measurement of parallel samples, which is common practice for this type of analysis. Therefore, we are unable to provide error bars for each data monitored in real time. We understand the importance of including error bars in the graphs to help assess the variability and reliability of the data. We regret that we are unable to provide error bars. However, after your reminder, to increase the transparency and credibility of the data, we have added the standard deviation to the mean data described in the text. We thank you for your understanding and look forward to your further guidance.

Line 214-215: "The average concentration of EPFRs was $(1.00\pm0.75)\times10^{14}$ spins/m$^3$ during NCP and $(8.19\pm5.60)\times10^{13}$ spins/m$^3$ during CP"

Line 255-258: "the average concentrations of G-ROS and P-ROS were $17.2\pm2.51$ nmol $H_2O_2$/m$^3$ and $13.6\pm2.71$ nmol $H_2O_2$/m$^3$, respectively, during NCP, decreased to $13.8\pm1.29$ nmol $H_2O_2$/m$^3$ and $7.25\pm1.79$ nmol $H_2O_2$/m$^3$ during period 2, and further decreased to $10.3\pm0.63$ nmol $H_2O_2$/m$^3$ and $7.02\pm0.57$ nmol $H_2O_2$/m$^3$ during period 3."

19. Line 194: Cite how you know that emissions primary combustion sources are significantly reduced. Is this specifically included in control measures?

**Response: Thank you for your question.** Based on the data presented in Section 3.1, we concluded that emissions from combustion sources experienced a significant reduction during control period. This is evidenced by the substantial decrease in the concentrations of primary pollutants, including $NO_2$, $SO_2$, and EC. The reductions of these pollutants are directly linked to the decreases in emissions from vehicles and industrial activities, both of which are combustion sources, indicating a significant reduction in emissions from combustion sources. As indicated in Table S2 of the Supplementary Material, these control measures include but are not limited to restrictions on motor vehicle usage and polluting industry restriction. The emission reduction effects of these measures are reflected in our monitoring data. With your reminder, we have revised it in Lines 194 and 243 in the manuscript to clearly demonstrate that the control measures have significantly reduced emissions from combustion sources.

Line 194: "Apparently, the control measures implemented during CP have effectively reduced emissions from industrial coal combustion and vehicle exhaust, both of which are important combustion sources."

Line 243: "The data presented above indicated that the generation of EPFRs with lower g-factor was decreased during CP when the emissions from combustion sources were significantly reduced."

20. Line 194: Grammatical suggestion: consider replacing "restricted" with "reduced" or "decreased" as restricted seems like it is contained within the control measures.

**Response: We are grateful for the suggestion.** We have replaced "restricted" with "decreased" in Line 243 of the revised manuscript as follows:

"The data presented above indicated that the generation of EPFRs with lower g-factor was decreased during CP when the emissions from combustion sources were significantly reduced."

21. Line 195: Possible misuse of "antioxidant properties." Please clarify why a more oxidized EPFR would have antioxidant properties. Do you mean instead not as easily oxidized compared to NCP?

**Response: Thanks to the reviewer for the correction.** There may have been a misunderstanding regarding use of "antioxidant properties" in the manuscript. You are right, we intended to express that EPFRs generated during CP are not easily further oxidized than EPFRs during NCP. The statement about "antioxidant properties" might have led to confusion, and we have corrected it in Line 245 as follow:

"Therefore, the free radicals generated during CP were less susceptible to further oxidation, while those generated during NCP were more easily oxidized."

22. Line 196: Clarify what is meant by "higher level" of Hp-p. Looking at the graph it appears the difference between the line width in strict CP and NCP falls within the same range, and the apparent difference may be due to the lesser amount of data points. Please comment on whether or not you think this may be the case.

**Response: Thank you for your comment.** We use the term "higher level" of $\Delta H_{\text{p-p}}$ in order to describe the increase in $\Delta H_{\text{p-p}}$ values during CP compared to NCP, but we neglected that it is not an accurate term for describing this increase. Additionally, after re-evaluating the data, we agree with your point that $\Delta H_{\text{p-p}}$ values during CP and NCP fall within the same range, and that the number of data points have an impact on the observed trend. Therefore, we have recalculated the average $\Delta H_{\text{p-p}}$ values during CP and NCP and conducted a comparative analysis. Thank you for your guidance, we have completely revised it in Line 246-248 as follows:

"The average $\Delta H_{p\text{-}p}$ of EPFRs during CP and NCP was $4.62 \pm 1.06$ G and $4.42 \pm 0.87$ G, respectively. The slightly larger $\Delta H_{p\text{-}p}$ during CP than NCP indicates a relatively complex path for the formation of EPFRs under strict control measure conditions."

23. Line 203-205: Include standard deviation for these measurements. Is the difference between them significant?

**Response: Thank you for your valuable suggestion.** We have provided the standard deviations of the G-ROS and P-ROS value in the manuscript. Furthermore, we have calculated the reduction percentages of G-ROS and P-ROS during CP. We found that the decrease in P-ROS concentration during CP compared to NCP was significant, and there was also a decreasing trend in G-ROS concentration. According to your suggestion, we have made the following modification in the manuscript:

Line 255-258: "the average concentrations of G-ROS and P-ROS were $17.2\pm2.51$ nmol $H_2O_2/m^3$ and $13.6\pm2.71$ nmol $H_2O_2/m^3$, respectively, during NCP, decreased to $13.8\pm1.29$ nmol $H_2O_2/m^3$ and $7.25\pm1.79$ nmol $H_2O_2/m^3$ during period 2, and further decreased to $10.3\pm0.63$ nmol $H_2O_2/m^3$ and $7.02\pm0.57$ nmol $H_2O_2/m^3$ during period 3."

Line 260-263: "Compared with NCP, the percentage decrease in G-ROS during CP was 24.1%, which was lower than that P-ROS decrease of 46.9%. This difference may be related to the complex formation and transformation mechanism of G-ROS. These results further suggested that the decrease in gaseous pollutants was lower than that in particulate pollutants."

24. Line 207: Clarify "much more," is that a factor of 2?

**Response: Thank you for your comment.** As suggested by reviewer 1, we added some discussions of the comparison of reduction effects in G-ROS and P-ROS during control period here. To ensure the compactness and focus of the manuscript, we have decided to remove this sentence "The percentage decrease in G-ROS and P-ROS concentrations during CP is much higher than that of EPFRs concentrations." from the manuscript. We hope it meets your satisfaction.

Line 259-263: "It is noteworthy that the impact of the control measures on G-ROS and P-ROS was different. Compared with NCP, the percentage decrease in G-ROS during CP was 24.1%, which was lower than that P-ROS decrease of 46.9%. This difference may be related to the complex formation and transformation mechanism of G-ROS. These results further suggested that the decrease in gaseous pollutants was lower than that in particulate pollutants."

25. Line 236: Please elaborate on this further. How does this compare to NO3 nighttime oxidation?

**Response: We are appreciative of the reviewer's suggestion.** After careful consideration, we have decided not to include a discussion on the relationship between EPFRs and nighttime $NO_3$ oxidation in the manuscript. The reason for this is that we lack data to support any association between EPFRs and nighttime $NO_3$ oxidation. Additionally, there is a dearth of research on the in-depth analysis of the relationship between EPFRs and nighttime $NO_3$ oxidation, thus the connection remains unclear. We look forward to exploring this area further in future study and appreciate your attention to this important scientific issue.

26. Line 284: The percentages are not in the correct order. Other sources are very significant (~30%) in the pie chart but only are listed as ~3%.

**Response: Sorry for our mistake.** The corresponding revision has been presented in Line 348 as follows:

"The percentage contributions of these source sectors to EPFRs during CP changed to 20.8%, 43.7%, 31.3%, 3.0%, and 1.25%, respectively."